# IMProofBench: Benchmarking AI on Research-Level Mathematical Proof Generation

## Abstract

As the mathematical capabilities of large language models (LLMs) improve, it becomes increasingly important to evaluate their performance on research-level tasks at the frontier of mathematical knowledge. However, existing benchmarks are limited, as they focus solely on final-answer questions or high-school competition problems. To address this gap, we introduce IMProofBench, a private benchmark consisting of $54$ peer-reviewed problems developed by expert mathematicians. Each problem requires a detailed proof and is paired with subproblems that have final answers, supporting both an evaluation of mathematical reasoning capabilities by human experts and a large-scale quantitative analysis through automated grading. Furthermore, unlike prior benchmarks, the evaluation setup simulates a realistic research environment: models operate in an agentic framework with tools like web search for literature review and mathematical software such as SageMath. Our results show that current LLMs can succeed at the more accessible research-level questions, but still encounter significant difficulties on more challenging problems. Quantitatively, GROK-4 achieves the highest accuracy of $61\%$ on final-answer subproblems, while GPT-5 obtains the best performance for proof generation, achieving a fully correct solution for $21\%$ of problems. IMProofBench will continue to evolve as a dynamic benchmark in collaboration with the mathematical community, ensuring its relevance for evaluating the next generation of LLMs.

## 1 Introduction

Large language models (LLMs) are making rapid progress on mathematical tasks, achieving strong results on challenging benchmarks like AIME (Balunovic et al., 2025) and FrontierMath (Glazer et al., 2024). These improvements suggest that LLMs may soon support mathematical research by collaborating with professional mathematicians on open problems. However, to determine whether current systems are capable of contributing in such settings, benchmarks are needed that test capabilities at the frontier of mathematical research.

**Limitations of existing benchmarks**  Existing benchmarks fall short of this objective: most focus on high-school or undergraduate-level mathematics (Balunovic et al., 2025; Frieder et al., 2023), due to the difficulty associated with designing rigorous, research-level problems. The few benchmarks that do target more advanced mathematics, like FrontierMath (Glazer et al., 2024) and HLE (Phan et al., 2025), focus exclusively on final-answer problems. As a result, they overlook proof-writing capabilities and allow models to apply shortcuts to reach the correct final answer without fully solving the problem (EpochAI, 2025).

**This work: IMProofBench**  To fill this gap, we introduce IMProofBench, a private benchmark developed in collaboration with the mathematical research community to evaluate LLMs on research-level proof writing. IMProofBench is built on a custom platform and supported by initiatives that actively involve professional mathematicians. It includes tasks ranging from challenging oral exam questions in a graduate course to open research questions based on the contributors' own work. Unlike static benchmarks, IMProofBench is designed as a platform for continuous evaluation: problems are added on a rolling basis, ensuring its continued relevance for evaluating the next generation of frontier LLMs. Currently, IMProofBench consists of $54$ problems developed in collaboration with over 35 mathematicians, with 29 more questions in the latest stages of the problem creation pipeline.

---

**Question: Isomorphism Classes of Stable Graphs**

Given an integer $g \geq 2$, let $N_g$ be the number of isomorphism classes of stable graphs of genus $g$ with precisely 3 edges. Give a closed formula for $N_g$ valid for all $g \geq 2$.
**Follow-up subquestions**: What are $N_3, N_8$, and $N_{10000}$?

| **GPT-5 Reasoning Summary** | **GROK-4 Reasoning Summary** |
|---|---|
| $N_g$ is a sum of connected multigraph types limited by 3 edges, considering partitions of genera. | Finds a non-closed formula for $N_g$, requiring to compute a sum from 1 to $g$. |
| 🐍 Obtain experimental data for some values. | 🔷 Attempts to look up the sequence in OEIS. |
| Submits a closed formula for $N_g$ as a degree-3 quasi-polynomial with a period of 6. | Submits a very concise sketch of the answer, with the non-closed formula for $N_g$. |
| **Full Grade**: 3/3    **Subquestions**: ✔✔✔ | **Full Grade**: 2/3    **Subquestions**: ✔✔✔ |

Figure 1: Example IMProofBench problem. Models are tested on research-level questions in an agentic framework with tool access. Grading of the main reasoning is done by a human expert, while follow-up subquestions are evaluated using an automated parser. For the question above, models other than GPT-5 and GROK-4 only made minor progress. For full details, see App. C.

**Problem creation pipeline**   Each problem in IMProofBench is authored by a research mathematician within their area of expertise. Submissions undergo a rigorous review process by a core team member and an additional mathematician with expertise in the relevant field. Reviewers provide feedback that allows authors to refine their problems before finalization. Alongside the main proof-writing tasks, authors are encouraged to add follow-up subquestions with final answers that can be automatically graded. These follow-ups enable a comparison between proof-writing and final-answer performance, while also supporting lower-cost evaluation across a broader range of models.

**Evaluation process**   Evaluation is conducted in an agentic framework designed to mirror a research environment. Models have access to computational tools such as Python and SageMath (sag, 2025), as well as web search and multi-turn reasoning. Each model is first tasked with solving the main problem, followed by the associated follow-up subquestions. The main solution is graded by the problem's author, who assigns a score from zero to three. Graders also annotate the types of errors, such as logical mistakes, and identify specific areas of partial progress, such as correct intermediate insights. Follow-up answers are automatically evaluated by comparing them with ground-truth solutions. We illustrate this process in Fig. 1, which shows a sample problem with two model solutions.

**Key results**   We evaluate 10 state-of-the-art LLMs on the current version of IMProofBench. Our results show that models can already solve a small but meaningful fraction of research-level problems: the best model, GPT-5, produces complete solutions for 22% of tasks, closely followed by GROK-4 at 19%. Notably, GROK-4 achieves the highest final-answer accuracy at 52%, surpassing GPT-5's 42%. Other models lag further behind, with CLAUDE-OPUS-4.1 scoring particularly poorly, only providing complete solutions in 3% of tasks.

**Qualitative analysis**   Beyond aggregate scores, our analysis reveals that many models are prone to reasoning errors, ranging from simple logical mistakes to deep misconceptions unlikely to be exhibited by any professional mathematician in the relevant area. Indeed, almost half of the model solutions contain arguments revealing fundamental misunderstandings of mathematical concepts, as judged by human graders. Moreover, models frequently hallucinate existing results to obtain a (flawed) answer. Finally, models almost never abstain from providing a solution attempt, preferring to present convincing but incorrect proofs rather than admit they are stuck. At the same time, they also show a wide-ranging familiarity with existing literature and can often provide insights that could meaningfully support mathematicians on a substantial number of problems. These results indicate that state-of-the-art LLMs can already aid mathematicians in their research, but also still need significant supervision to avoid simple mistakes.

**Core contributions**   The core contributions of this work are:

- IMProofBench, a private and evolving benchmark for research-level problems, developed in collaboration with the mathematical community.

- A systematic analysis of proof generation capabilities across state-of-the-art LLMs, demonstrating that GPT-5 provides fully justified solutions for a small but non-trivial fraction of our research-level problems, as judged by human expert graders.

- A qualitative analysis discussing both the difficulties and strengths of current state-of-the-art models and their potential application to research-level mathematics.

## 2   RELATED WORK

We briefly review existing benchmarks that evaluate LLMs on mathematical reasoning tasks.

**High-school and undergraduate benchmarks**   High-school and undergraduate problems are the most common source of mathematical benchmarks due to their wide availability. Examples include GSM8k (Cobbe et al., 2021) and MATH (Hendrycks et al., 2021), along with more recent efforts such as OmniMath (Gao et al., 2024), UGMathBench (Xu et al., 2025), and MathArena (Balunovic et al., 2025). However, these benchmarks fail to measure model performance on realistic, research-level tasks. Furthermore, even the most challenging competition problems are increasingly tractable for state-of-the-art LLMs (Balunovic et al., 2025), meaning that these benchmarks are reaching their saturation point. More importantly, most of these benchmarks do not evaluate proof-based reasoning.

**Research-level benchmarks**   To move beyond competition problems, several benchmarks aim to capture research-level mathematical reasoning, though each has notable limitations, and none provide systematic proof evaluation. FrontierMath (Glazer et al., 2024) offers extremely challenging private problems, though privileged access by OpenAI raises concerns about evaluation fairness (AI, 2025). Humanity's Last Exam (Phan et al., 2025) crowd-sources expert-level questions across domains, including mathematics, but suffers from contamination risks due to its open nature and reports of substantial noise in the benchmark (Skarlinski et al., 2025). RealMath (Zhang et al., 2025) sources problems from arXiv papers, enabling dynamic evaluation of research-level problems, but it is currently not being maintained. Finally, the UQ-Dataset (Nie et al., 2025) collects unsolved StackExchange questions, many of them mathematical. While promising, it lacks systematic human evaluation of proof validity, making consistent cross-model comparisons difficult.

**Proof-based benchmarking efforts**   The importance of evaluating proof-generation capabilities has recently gained attention, leading to a range of benchmarking efforts. For example, Mahdavi et al. (2025) showed that models trained with reinforcement-style methods such as GRPO (Shao et al., 2024) perform poorly at proof writing. However, more recent evaluations on the USAMO and IMO 2025 demonstrated substantial progress in the ability of frontier models to construct rigorous mathematical arguments (Petrov et al., 2025; Balunovic et al., 2025). At the same time, other studies highlighted a persistent gap between final-answer accuracy and genuine proof-writing ability, indicating that final-answer benchmarks are not sufficient to measure mathematical capabilities (Guo et al., 2025; Dekoninck et al., 2025). Despite these advances, benchmarks remain focused on high-school and undergraduate mathematics, leaving research-level proof generation unexplored.

**Formal math benchmarks**   A complementary line of work evaluates LLMs on their ability to generate proofs in formal systems such as Lean (de Moura and Ullrich, 2021). Success in this setting typically requires fine-tuning frontier models for this particular task (Ren et al., 2025; Lin et al., 2025), as off-the-shelf LLMs perform poorly. Formal proofs offer the advantage of automatic verification and scalable evaluation, but current models still lag significantly behind their natural language proof counterparts (Dekoninck et al., 2025). Benchmarks in this space include PutnamBench (Tsoukalas et al., 2024) and MiniF2F (Zheng et al., 2022), which formalize problems from well-known mathematics competitions into Lean or Isabelle. Work on a Lean-based benchmark with research-level problems is currently in progress with the *ProofBench* initiative (Bowler and Carmesin).

Figure 2: Workflow for question creation with peer review. Authors iteratively refine questions based on expert review. A problem is only accepted once the reviewers have no further comments.

## 3 BENCHMARK METHODOLOGY

In this section, we present the creation and evaluation process of IMProofBench. We begin by outlining our community outreach efforts (§3.1), followed by a description of the problem creation pipeline (§3.2) and the evaluation methodology (§3.3). Finally, we discuss the current state of the benchmark and our plans to maintain and extend it as a platform for continuous evaluation (§3.4).

### 3.1 COMMUNITY OUTREACH

Creating a novel and diverse collection of research-level problems is a challenging task, requiring professional mathematicians from a wide range of fields. To facilitate this, we undertook several initiatives to engage the community:

- **Workshops**: We organized several problem-creation sessions as satellite events at mathematical research conferences.
- **Posters and flyers**: We distributed informational materials in math common rooms and conference venues to reach graduate students, postdocs, and faculty.
- **Personal outreach**: Organizers and motivated contributors actively contacted their academic networks to invite participation.

These efforts are ongoing as we continue to expand the benchmark. Informal surveys of contributors indicate that key motivating factors to participate include convenient access to frontier models via the platform, curiosity about AI-generated responses to submitted questions, and the opportunity for co-authorship on resulting publications for contributors whose questions are accepted.

### 3.2 PROBLEM CREATION PIPELINE

**Question creation** As shown in Fig. 2, authors draft questions through a dedicated web interface and can immediately test them on an instance of GPT-5 configured with high reasoning effort, built-in web search and code interpreter tools, and safeguards such as a 30-minute timeout and a cap of 20 evaluations per day to prevent abuse. This LLM interaction allows quick, optional feedback on both difficulty and potential ambiguities. Importantly, problem selection criteria are independent of the model's performance on the draft question. Where possible, authors are asked to include follow-up subquestions with unique, automatically gradable answers, with the option to assign point weights for the solution of different subquestions to reflect their difficulty or importance. This facilitates broader evaluation of more models by reducing reliance on human grading, while also supporting comparisons between final-answer accuracy and proof-generation capability. To guide contributions, authors receive detailed instructions that include illustrative examples and emphasize that questions should require PhD-level insight, while avoiding standard textbook exercises or computational problems. A complete description of the author instructions is provided in App. B.2.

**Question peer-review process** Once a question is submitted, an administrator recruits a reviewer whose expertise aligns with the problem's subject area. Reviewers are invited via email, with invitations extended to both existing benchmark participants and external experts if necessary. The review process follows an academic peer-review model, with the administrator and reviewer providing

Figure 3: Evaluation workflow in a multi-turn environment with research tools. The main solution is graded by a human expert, while follow-up questions are automatically evaluated.

detailed feedback, asking for revisions where necessary. While the reviewer concentrates on verifying mathematical correctness and difficulty, the administrator ensures that the submission adheres to the guidelines. Authors are then invited to revise their problem and respond to comments with clarifications or adjustments. A problem is accepted only after both the administrator and reviewer have no remaining concerns. A full description of the reviewer instructions is given in App. B.3.

## 3.3 MODEL EVALUATION

**Evaluation environment** As shown in Fig. 3, models are evaluated within an agentic framework designed to approximate real research conditions. We use the Inspect framework (AI Security Institute, 2024) and give models access to a diverse set of tools:

- **Python**: a full scientific environment with NumPy, SciPy, SymPy, and related libraries.
- **Bash**: an Arch Linux console with persistent filesystem and computer algebra systems like GAP (GAP, 2024), and Maxima (Maxima, 2025).
- **SageMath**: open-source mathematical software with specialized packages and mathematical databases (sag, 2025).
- **Web search**: a tool for retrieving literature and external references.

A full description of these tools is provided in App. E. To submit an answer, models must use a dedicated submit tool, which ensures a clear distinction between intermediate reasoning steps and the final output. The submitted answer is either presented to the human grader for main questions or compared with ground-truth answers for follow-up subquestions. Each model is allocated up to 300,000 tokens for main questions, with an additional 100,000 tokens available for each follow-up, supporting extended interaction and tool use. In App. I.1 we describe some ablation experiments where models were tested with a simpler one-turn evaluation and restricted tools. While some models like GROK-4 show significant performance drops in this restricted setting, others like GPT-5 show more complex patterns, exhibiting *stronger* performance with fewer tools.

**Model selection and tiers** To ensure scalability, we adopt a tiered evaluation system. Each model is assigned to a tier that reflects its priority for human grading, allowing question authors to focus on the most important submissions when their time is limited. The highest-priority tier includes state-of-the-art models that demonstrate strong performance on existing benchmarks: GPT-5 (OpenAI, 2025b), GEMINI-2.5-PRO (DeepMind, 2025), GROK-4 (xAI, 2025), and CLAUDE-OPUS-4.1 (Anthropic, 2025a). Lower tiers currently include O3 and O4-MINI (OpenAI, 2025a), GPT-4O (OpenAI, 2024), GEMINI-2.5-FLASH (DeepMind, 2025), GROK-3 (xAI, 2025), and CLAUDE SONNET 4 (Anthropic, 2025b). A complete description of the tiers is provided in App. D.

**Grading process** Scoring of model answers takes place in two separate stages. First, follow-up subquestions are automatically graded by comparing the model's output with the ground-truth reference. Currently, this automated evaluation is also manually verified by an administrator, who can correct parsing errors and update the grading script if necessary. In the second stage, human grading is conducted through our dedicated web interface. The question's author serves as grader and provides three types of feedback:

- **Error classification**: identifying reasoning mistakes caused by incorrect logic, hallucinations, calculation errors, or conceptual misunderstandings.
- **Achievement indicators**: marking whether the model demonstrated understanding, reached correct conclusions, identified key insights, or produced useful reasoning.
- **Overall progress**: assigning a score of no (0/3), minor (1/3), major (2/3), or full (3/3) progress.

Error classification and achievement indicators are recorded as eight binary marks and enable a more fine-grained analysis of model performance. In particular, this structure allows us to identify both the areas where models can already assist research mathematicians and the areas where they remain most prone to errors. To avoid bias, model identities remain hidden until grading is completed.

In App. I.2 we describe results from a preliminary analysis verifying grader reliability. While the agreement on overall progress score is quite good (89% of pairwise comparisons within one point of each other), the individual error and achievement categories show more mixed results (ranging from 84.9% for Correct End Results, to only 58.2% for the Incorrect Logic indicator). We do note that for now the size of the comparison group is still small, and that we plan to conduct more comprehensive grading comparisons in the future and study the sources of disagreement.

### 3.4 BENCHMARK STATISTICS AND FUTURE DEVELOPMENT

**State of the benchmark**   IMProofBench is under active development, with this paper presenting its first pilot phase. This initial version consists of 54 questions and 120 follow-up subquestions. Topics range from areas of pure mathematics, such as algebraic geometry, combinatorics, and graph theory, to applied subjects such as stochastic analysis and bioinformatics. In Fig. 13 of App. A, we include a word cloud of question tags, weighted by frequency. Of the 54 benchmark problems, authors characterize 13 as open research questions. A total of 35 mathematical researchers have contributed at least one question in their area of expertise.

**Continuous development**   With models showing rapid progress in mathematics, benchmarks are being saturated at an accelerating pace. For example, the USAMO 2025 benchmark moved from a solve rate below 5% to more than 60% in only a few months (Petrov et al., 2025; xAI, 2025). To ensure that IMProofBench remains both unsaturated and challenging, we are committed to its continuous development along several dimensions. First, we will maintain our problem creation pipeline and accept problems on a rolling basis, while forming new strategic partnerships with leading mathematical institutions to keep problem difficulty aligned with the capabilities of future models. Second, to prevent contamination, we will employ a dynamic problem management system in which authors are encouraged to revisit and possibly retire their problems once new publications or techniques make them significantly easier. Third, we plan to create a transparent interface that allows major companies or research labs to configure and provide their own agents for solving problems in IMProofBench. This creates an opportunity to provide objective and equitable evaluations of internal research models, ensuring that the benchmark reflects the latest state-of-the-art models and agents in realistic settings. Other ideas for future work are given in App. F.

## 4 EXPERIMENTAL RESULTS

In this section, we give quantitative and qualitative summaries of model performance on IMProof-Bench. In §4.1, we compare final-answer correctness and proof-generation capabilities of several frontier LLMs. Then, in §4.2, we present a detailed analysis of errors and achievements made by these models. In §4.3, we analyze token and tool usage. We conclude in §4.4 with a qualitative discussion of several notable examples and overall results.

### 4.1 MAIN RESULTS

**Proof-based evaluation**   As illustrated in Fig. 5, GPT-5 achieves the strongest performance, producing a complete solution in 21% of cases. It fails to make any progress on only 17% of the questions, showing that the model can engage meaningfully with most problems in the benchmark. These results highlight both the impressive capabilities of current systems and the difficulty of IMProofBench, as substantial progress remains possible. Importantly, none of the 13 open problems were solved.

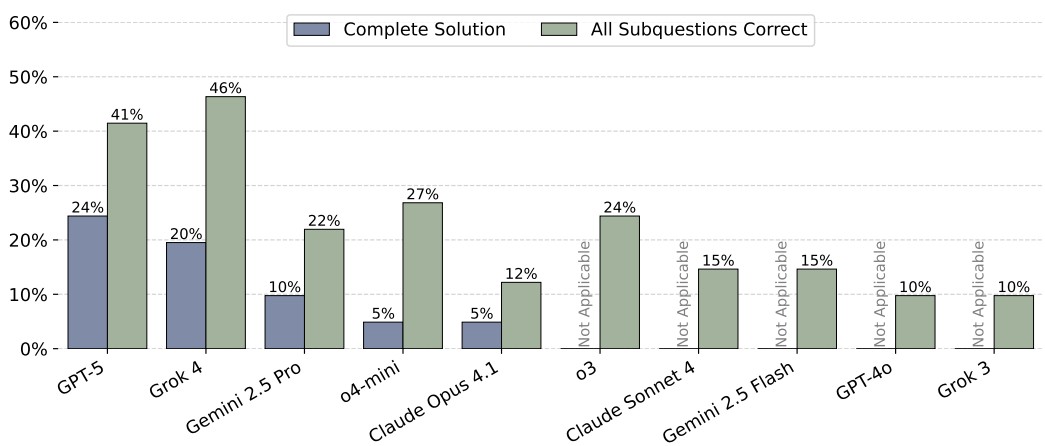

Figure 4: Results on the 41 questions that include follow-up subquestions and human grading.

**Final-answer evaluation** In Fig. 4, we compare performance between final-answer accuracy and full proof-based evaluation on the 41 questions that include both. While GROK-4 is slightly worse compared to GPT-5 on proof-based evaluation, it performs best on final-answer accuracy, obtaining 52%. The ranking of other models is consistent between the two evaluation modes. Furthermore, the correlation coefficient between author-weighted subquestion scores and the 0–3 progress scores assigned by human graders is 0.48, while the Pearson correlation between getting *all* subquestions right and a full human grade of 3 for the main ques-

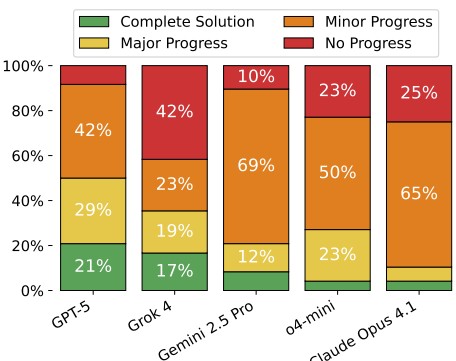

Figure 5: Results on IMProofBench.

tion is 0.43. This suggests that final-answer evaluation is a useful proxy for model ability, but human grading provides essential nuance and a more refined view of performance. In App. A, we analyze final-answer accuracy over the full set of questions, including partial progress.

## 4.2 ERROR AND PROGRESS ANALYSIS

We now analyze the error and achievement indicators classified by the question authors. This provides a clearer picture of where models fail and where they already provide meaningful help.

**Error indicators** As shown in Fig. 6, models make a wide variety of errors. Logical errors are the most common, with models frequently introducing unfounded assumptions or claiming incorrect implications. CLAUDE-OPUS-4.1 is particularly weak in this respect, having logical errors in nearly 80% of its responses. Conceptual errors are also widespread. Importantly, these errors are described as fundamental misunderstandings of mathematical concepts in the grader guidelines, showing that models do not fully understand some advanced mathematical concepts. Furthermore, hallucinations are surprisingly frequent, with GEMINI-2.5-PRO hallucinating results in 50% of its answers. In contrast, calculation mistakes are rare, which is expected since problems are proof-oriented and models can rely on tools to perform calculations. A notable outlier among all models is GROK-4: it often produces extremely short answers that only contain a final answer attempt without supporting arguments. This leads graders to be unsure about the precise mistakes or achievements in its reasoning.

**Achievement indicators** As shown in Fig. 7, most models demonstrate general familiarity with the background knowledge needed to understand the problems, which is an impressive achievement given that many of these questions reference highly specialized mathematical concepts. Creative ideas are rarer, but GPT-5 still displays non-trivial creativity in almost half its solutions. This indicates that the model can already make remarkable progress on difficult problems. Finally, in some cases, models

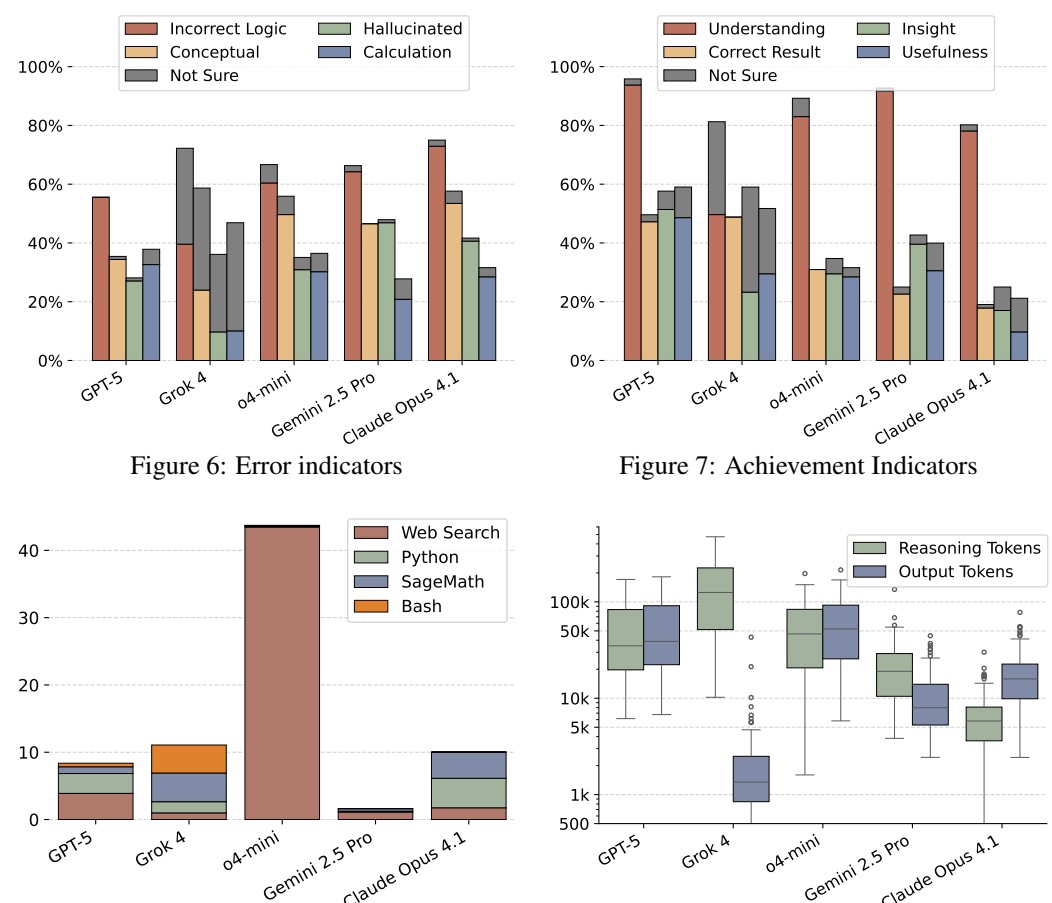

Figure 6: Error indicators

Figure 7: Achievement Indicators

Figure 8: Average tool usage per question.

Figure 9: Token usage per question.

provide insights that could be helpful to expert mathematicians, with GPT-5 offering meaningful contributions in about half its attempts. This is a significant achievement for any automated system.

### 4.3 TOOL AND TOKEN USAGE

As illustrated in Fig. 8 and Fig. 9, models vary widely in their resource usage, both in tool selection and token consumption. GROK-4 spends almost three times as many reasoning tokens as other models while producing relatively few output tokens. It is also the only model to make heavy use of the bash tool. Inspection of its logs shows frequent use of the command line to download research papers from arXiv (via `wget` or `curl`) and to convert them using utilities like `strings` or `gs`. This sometimes gives GROK-4 an edge over models that rely only on internal search tools. Another pattern is that O4-MINI relies heavily on the web search tool, averaging over 40 searches per problem, while CLAUDE-OPUS-4.1 makes frequent use of Python, occasionally misusing it as a scratchpad with many comments or static `print` statements. Usage plots for all models, including those only evaluated on final-answer questions, are shown in App. A.

### 4.4 QUALITATIVE ANALYSIS

We now describe qualitative observations drawn from manual inspection of logs and grader comments.

**Broad and deep literature knowledge** Leading models such as GPT-5 show strong familiarity with the mathematical literature and are often able to identify specialized results in published work. However, they struggle to locate more obscure sources, such as private lecture notes, which human experts often use.

**Use of specialized tools**   When confronted with complex computations, models frequently employ tools like SageMath. However, more specialized packages that are accessible through the bash tool pose challenges, with models often producing syntactically invalid code. After repeated failures, they sometimes revert to more common libraries, e.g., by using a manual Python re-implementation.

**Mistakes are often hidden**   Models are typically quite economical with their mistakes, adding just a single simplifying assumption or incorrect claim. This one mistake often makes the problem significantly easier but leads to incorrect conclusions. Importantly, they are usually presented with confidence and framed rhetorically, for example, by stating that a "well-known result" implies a key step. Sometimes, different models even independently converge on the *same* shortcut, leading to parallel arguments that can create a false sense of consensus for the user. Although reasoning traces are often not accessible, we did not find evidence of *deliberate deception* where models were aware of their own mistakes and presented the flawed argument nonetheless.

**Models rarely abstain**   Models rarely abstain from claiming a solution to the presented IMProof-Bench questions. Even on the extremely challenging open problems in the benchmark, models almost always make an attempt at a definite answer. This happens despite user preferences strongly favoring an abstention over a mistaken but convincing proof.

**GROK-4 gives short responses**   As noted earlier, GROK-4 often provides only a final answer, particularly when the question allows for a short response. This occurs despite repeated instructions to provide full proofs (see App. H). Combined with the hidden reasoning tokens in the GROK-4 API, this made evaluations difficult and led to frequent "Not Sure" grades on our binary categories.

**User testimonials**   For many contributors, this benchmark was their first hands-on experience with state-of-the-art LLMs in an agentic setup. Participants at outreach events expressed surprise at the level of performance (*"Quite impressive, especially the case of degree 3 where one has to argue a little bit..."*). During grading, we found that some models applied new approaches to known problems, surprising the expert graders (*"Interestingly, I was not familiar with the correct solution from the models, even though it is relatively fundamental."*). Although no open problems were solved, some attempts received positive feedback (*"Still I am amazed by the quality of the one-shot answers."*).

## 5   LIMITATIONS

The main limitation of IMProofBench is its current scale, with only 54 questions included so far. However, we are continuously expanding the benchmark, with an additional 14 problems at an advanced draft stage and 29 problems in the final stages of review. Even at this point, our analysis already provides detailed and valuable insights into the potential of LLMs for research-level mathematics, and these findings will become even more compelling as the benchmark develops further. Much smaller-scale evaluations of proof-based problems, such as those conducted on the USAMO and IMO 2025 (Petrov et al., 2025; Balunovic et al., 2025), have already produced meaningful conclusions, which underscores the value of such efforts even when the number of problems is small.

## 6   CONCLUSION

In this paper, we introduced IMProofBench, a benchmark designed to evaluate research-level proof-writing capabilities in LLMs. Unlike prior datasets that focus primarily on final answers, IMProof-Bench evaluates whether models can produce logically sound arguments that meet the standards of mathematical research. Each problem is authored and peer-reviewed by professional mathematicians, and evaluation takes place in an agentic framework that mirrors a real research environment. Our experiments with state-of-the-art LLMs show that models can already solve a meaningful subset of research-level problems, with GPT-5 achieving complete solutions on 22% of tasks. These findings highlight that while current models remain imperfect and prone to errors, they are already capable of providing valuable support to working mathematicians for some problems.

## REPRODUCIBILITY STATEMENT

While the IMProofBench dataset remains private, we take several measures to ensure transparency of the resulting evaluations: we give detailed descriptions of tested models and their API configuration (App. D), an account of the tools available to them within the Inspect framework (App. E), and the used evaluation prompts (App. H). We plan to release the code base of our web platform and evaluation framework under a suitable open-source license before November 30, 2025, and to actively encourage scrutiny, feedback, and participation by outside developers. This release will include a continuously expanding collection of open sample problems that allow users to test the relevant systems and reproduce our data analysis on this sample set.

Moreover, we are open to scientific collaborations with outside parties to conduct specific investigations using our problem and grading dataset. Such requests will be evaluated on a case-by-case basis with the aim of ensuring the privacy commitments we make to our contributors.

## ETHICS STATEMENT

We acknowledge that our project received support in the form of free credits for both the xAI and the Gemini APIs, for which we thank the teams at the respective companies. These contributions did not have an influence on our scientific evaluation of the respective models, which happens via a model-agnostic framework.

We address several further ethical considerations:

- **Contributor protection**: Problems remain private to protect contributors' intellectual property, with generous withdrawal policies if questions lead to publishable insights. Contributors maintain rights to their content and receive co-authorship on benchmark publications.
- **Responsible AI evaluation**: By keeping the dataset private and focusing on evaluation rather than training data provision, we aim to measure capabilities without directly improving them.

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

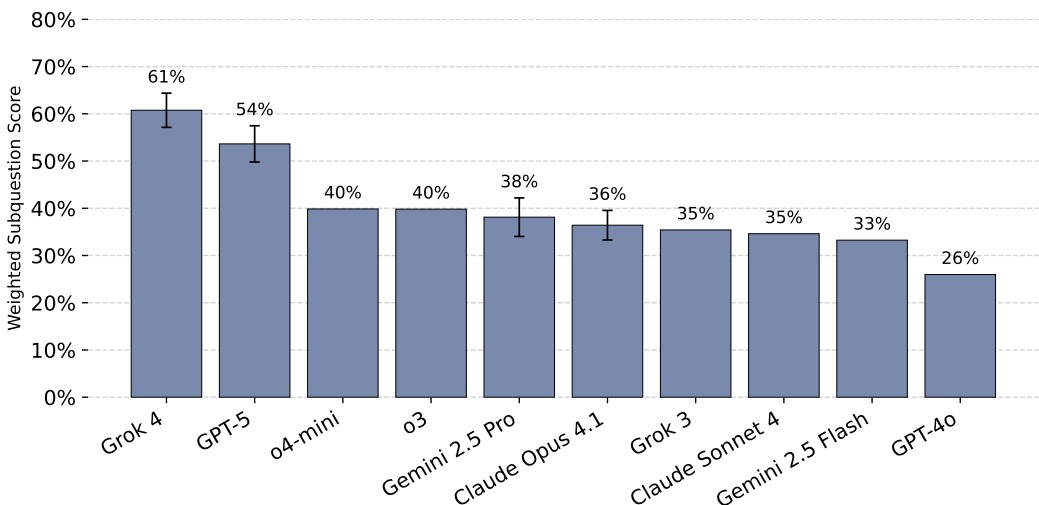

Figure 10: Average percentage of points for subquestion evaluation. Here, performance on any individual question is weighted by the point rewards determined by the problem author. For the models with at least two evaluations per question, we plot error bars (as explained in I.3).

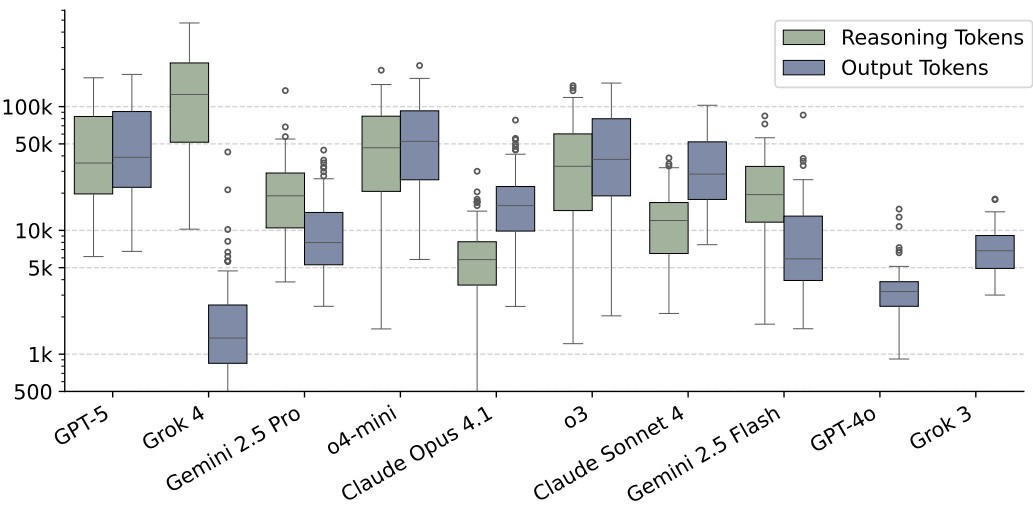

Figure 11: Token usage distribution for problem evaluation (main question and subquestions) for all tested models.

## A  BENCHMARK COMPOSITION AND ADDITIONAL EVALUATION RESULTS

**Performance on final-answer subquestions**   In Fig. 10, we present the average scores obtained by all 10 evaluated models on the final-answer subquestions, using the author-appointed weights that reflect importance or difficulty. As shown in the figure, GROK-4 achieves the highest performance, with almost a 10% margin over the second-ranked model, GPT-5.

**Token usage**   In Fig. 11, we show the distribution of reasoning and output tokens across the evaluated questions. GROK-4 produces the longest reasoning traces but the shortest outputs among all models in the benchmark, consistent with the trend described in §4.4. In contrast, the OpenAI models show a more balanced ratio of reasoning to output tokens. The Gemini models use slightly more reasoning tokens, while the Claude models generate more verbose outputs. With respect to

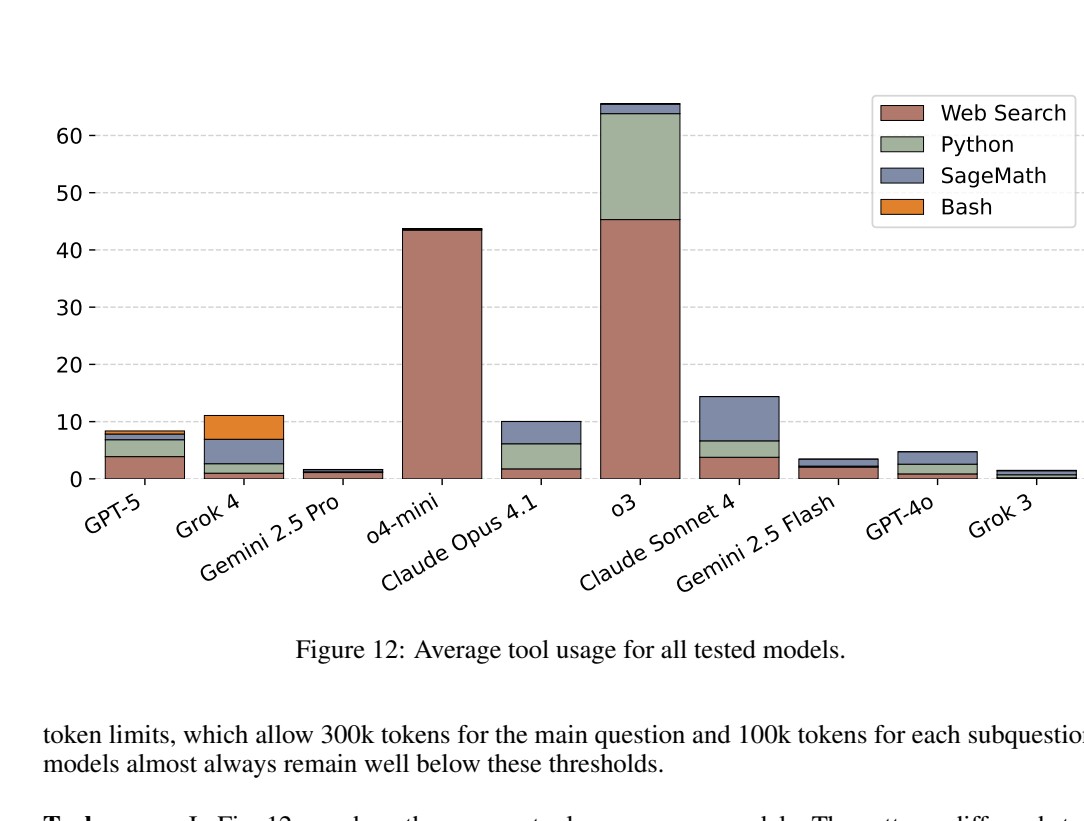

Figure 12: Average tool usage for all tested models.

token limits, which allow 300k tokens for the main question and 100k tokens for each subquestion, models almost always remain well below these thresholds.

**Tool usage**  In Fig. 12, we show the average tool usage across models. The patterns differ substantially. O4-MINI and O3 each make around 50 tool calls per problem, relying more heavily on the web search tool than any other model in the leaderboard. Further, GROK-4 is the only model that makes frequent use of the bash tool. Other models display broadly similar usage patterns, distributing their calls among web search, Python, and SageMath.

**Topics in IMProofBench**  In Fig. 13, we display the distribution of problem tags in IMProofBench. The topic of "Algebraic Geometry" currently dominates, reflecting the research focus of the benchmark organizers. These organizers both contributed problems themselves and solicited input primarily from colleagues in their own academic networks. Future development of the benchmark will aim to broaden its coverage to include a wider range of topics in pure and applied mathematics, as outlined in App. F.

## B  HUMAN INTERFACE AND INSTRUCTIONS

In this appendix, we discuss how contributors and benchmark administrators interact with IMProof-Bench, including the instructions and interface for different steps of the submission process (question generation, review, and grading). In App. B.1, we give a brief overview of the main pages on the web interface. Then, in App. B.2, we provide details on how questions are created and edited. In App. B.3, we explain the review process. Finally, in App. B.4, we discuss the grading interface.

### B.1  SUBMISSION WEBSITE

Contributors submit problems via a secure website designed for submitting and reviewing questions, and grading AI answers (see Fig. 14). Features include:

- **User accounts and permissions**: Contributors can create an account tied to a (verified) email, which allows them to author questions and use website features like the free AI solution previews for these questions. Benchmark administrators have additional access to manage model evaluations, review requests, and access a live view of benchmark results.
- **Community features**: The website shows a list of contributors (ordered by the number of accepted questions or similar parameters) to encourage active participation, and links to a project Zulip with further news and an opportunity to provide feedback.

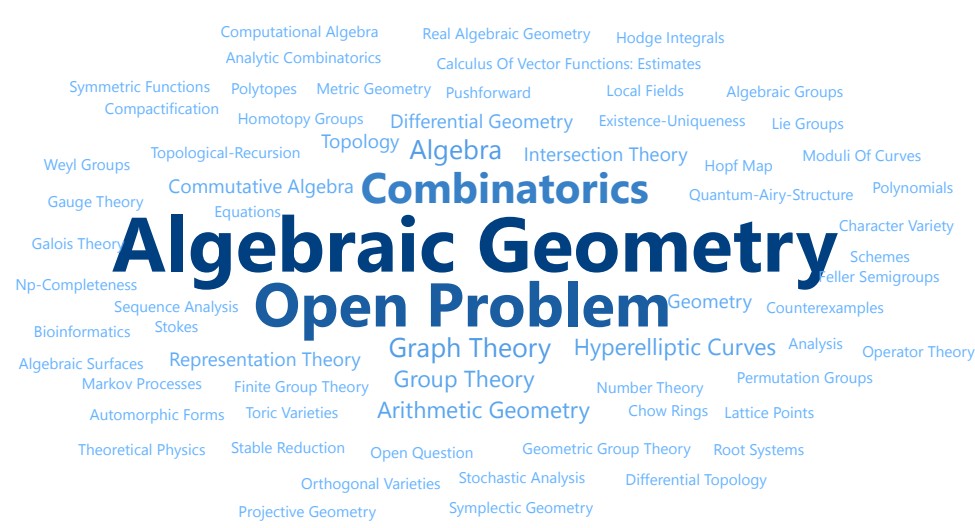

Figure 13: Word cloud of tags assigned to IMProofBench problems.

- **Benchmark dashboard**: Total numbers of contributors and questions in different stages of the submission process are displayed to show project progress. An overview page with both live results and archived snapshots of the benchmark state will be added in the future.
- **About the project**: Information about the IMProofBench is provided. This information contains the initial whitepaper, an overview of core team members, a timeline of planned steps, and a page with frequently asked questions. A privacy policy detailing our handling of user data is linked in the footer of the page.

### B.2 QUESTION CREATION AND EDITING

Benchmark problems are created through a structured interface that guides contributors through the submission requirements. The system provides comprehensive guidelines (see Figure 15) emphasizing the key characteristics of suitable benchmark problems.

**Problem guidelines**    Effective benchmark problems must meet several criteria:

- **PhD-level difficulty:** Problems should be suitable for oral exams of graduate courses, research papers, or advanced seminars, representing mathematics close to or at research-level.
- **Genuine mathematical insight:** Solutions must require non-routine approaches that cannot be solved through pattern matching or standard algorithm application.
- **Clear proof-based main question:** The primary answer should consist of a complete mathematical argument rather than merely a numerical result.
- **Auto-gradable subquestions:** Each problem requires 2–3 subquestions with unique answers (e.g., "Is the statement true for $n = 5$?" or "What is the rank of this group?"), enabling automated evaluation.

Contributors should avoid problems solvable by lucky guessing, standard textbook exercises (even from graduate texts), or purely computational problems that mathematical software can solve directly.

**Question editing interface**  The question creation and editing window (see Figure 16) provides a comprehensive authoring environment with the following components:

- **Main question editor:** A text area supporting Markdown with LaTeX mathematics, featuring a live preview pane that renders the formatted content in real-time. Contributors can use standard LaTeX delimiters (`$...$` for inline and `$$...$$` for display mathematics).
- **Problem metadata:** A tags field allows contributors to categorize problems by area (e.g., "group theory", "representation theory", or "permutation groups") and special characteristics (e.g., "open problem" for questions where the author seeks but does not know the answer).
- **AI solution preview:** Contributors can test their questions against a frontier AI model (currently GPT-5 with high reasoning effort) using up to 20 free attempts per day. This feature helps authors evaluate whether their problem has appropriate difficulty and clarity.
- **Sample solution:** A dedicated editor for the complete solution, which serves as the reference for reviewers and graders. The solution should demonstrate the expected level of rigor and detail to allow expert review to verify correctness and serve as a reference for grading model answers.
- **Subquestions management:** A dynamic form system for adding multiple subquestions, where each subquestion consists of:
    - Question text (supporting Markdown and LaTeX)
    - Expected answer field for the unique answer
    - Evaluation method selector (e.g., exact match)
    - Optional points value (defaulting to 1) for weighting subquestions by difficulty or importance
    - Rationale field for explaining the correct answer

**Question detail view**  Once submitted, questions are displayed in a detail view (see Figure 17) that presents all components in their rendered form. This view shows:

- The question status in the submission pipeline (Draft → Under Review → Approved → Active)
- Rendered the main question and sample solution with properly formatted mathematics
- List of subquestions with their expected answers
- AI solution attempt preview when available
- Review comments from expert reviewers (when in review stage)
- Response interface allowing authors to address reviewer feedback and revise their submission

The detail view serves as the central hub for tracking a question's progress through the review process and facilitating communication between authors and reviewers.

### B.3 REVIEW PROCESS AND INSTRUCTIONS

Each question is reviewed by at least one expert before being included in the benchmark. These experts are invited to submit a review via email. An example of such an email is included below.

---

**Reviewer invitation email**

```
Dear [invited_user],

My name is [inviting_user] and I am part of a small team of mathematicians studying the
question of how good today's AI models are at solving research-level math questions. As
part of this IMProofBench project, we are building a collection of challenging
mathematical problems to use for testing the AI performance.

We would like to ask for your help in verifying the mathematical correctness of one such
question. If you are interested to learn more about the project, further information is
available at https://improofbench.math.ethz.ch/faq/

The following question was submitted for inclusion in the IMProofBench dataset:

Title: Permutation representation
Author: Example Participant
```

---

```
Would you be willing to review this question and:
- Verify that the phrasing is well-defined and unambiguous
- Confirm the provided solution is mathematically correct
- Make any suggestions for improvements (e.g., additional unique-answer subquestions)

We estimate that for most problems, this should take between 10 and 30 minutes.

You can view the full submitted problem and write a review at:
[ACCEPT_URL]

There, you will also have the option to decline this review request after viewing the
question.
Alternatively, you can decline immediately by clicking:
[DECLINE_URL]

If you provide a review, the question's author will be notified and have the chance to
revise the question and compose a response. After seeing the response, you have the
option to submit a further review or recommend the question for acceptance in the
benchmark.

Thank you for considering this request!

Best regards,
[inviting_user]

Note: To track your review and allow you to see the author's replies, accepting the
review request will create a user account for you on our website. You can optionally set
a password after submitting your review to log back in and e.g., contribute a question to
 the benchmark yourself.
```

When the reviewer accepts the review invitation by clicking on the link, they are forwarded to a webpage displaying the problem to be reviewed, along with a form for review submission and further information (see Figure 18). The reviewer may also view the full review guidelines displayed in Figure 19. The review consists of a short comment by the reviewer indicating improvements and/or mistakes in the question statement. Before submitting the review, the reviewer decides on a recommended action among the following: "Recommended for acceptance", "Needs revision" and "Not suitable". The site admins are notified when a review is complete and can take action accordingly. If the reviewer selects "Not suitable", the question is automatically reset to the "draft" status. Independent of the outcome, the author is permitted to submit an answer to the reviewer's comments and change the question if necessary. The reviewer may then either submit a new review taking into account the changes, or a new reviewer may be invited.

### B.4 GRADING INTERFACES

The grading system provides a structured interface for human evaluation of model-generated proofs through a dedicated web page.

**Human grading interface**  The main grading interface (see Figure 20) employs a three-column layout designed to facilitate easy access to relevant information and the feedback form:

- **Left column:** Displays the question statement and sample solution for reference
- **Center column:** Shows the model's complete response with mathematical rendering
- **Right column:** Contains the interactive grading panel with scoring controls

To prevent bias, model identities are concealed behind randomized aliases (Answer A, B, C, etc.) that remain hidden until all answers for a question have been graded. The system maintains independent grading sessions for each evaluator, with aliases shuffled differently to ensure blind evaluation.

**Grading categories**  The scoring form consists of three main components providing multifaceted evaluation, with relevant information available via concise tooltips:

**AI mistake indicators:** Four binary categories identifying common failure modes:

1. **Incorrect Logic:** Flawed logical steps or invalid reasoning
2. **Hallucinated:** References to non-existent theorems, papers, or results
3. **Calculation:** Arithmetic or algebraic errors

4. **Conceptual:** Fundamental misunderstanding of mathematical concepts

**AI achievement indicators** Four binary categories recognizing positive aspects:

5. **Understanding:** Correctly identifies what needs to be proven or calculated
6. **Correct Result:** Arrives at the correct final answer (with N/A option for open-ended problems or when the correct answer is unknown)
7. **Insight:** Shows creative problem-solving or novel approaches
8. **Usefulness:** Solution would be helpful to someone learning this topic

Each binary category offers three response options: "True", "False", or "Not Sure", allowing graders to indicate uncertainty when evaluation is ambiguous.

**Overall progress** A four-point scale (0–3) rating overall solution progress:

- **0/3:** No progress toward solution
- **1/3:** Minor progress with limited advancement
- **2/3:** Major progress with substantial work completed
- **3/3:** Complete solution achieved

This overall progress score serves as the primary metric for model ranking and comparison.

**Additional grading features** The interface includes several supporting elements to ensure grading consistency and quality:

- **Grading notes:** A persistent text area where graders record their evaluation criteria and decision patterns across all answers (e.g., "Matrix errors count as Calculation, Theory errors as Logic"). These notes help maintain consistency when grading multiple model responses and facilitate reproducibility in future grading sessions.
- **Comments field:** Answer-specific observations about edge cases or explanations for grading decisions.
- **Auto-save functionality:** Grading selections are automatically preserved with a 2-second debounce to prevent data loss.
- **Focus mode:** An optional distraction-free interface that maximizes screen space by hiding navigation elements and allowing collapsible panels, enabling graders to concentrate on detailed evaluation.
- **Flag for organizers:** Option to mark responses requiring special attention due to serious issues or technical problems.

The grading workflow supports iterative evaluation, allowing graders to mark answers as complete, incomplete, or given up (for responses that cannot be meaningfully evaluated). Once all model answers for a question are marked complete, the system reveals the true model identities, enabling post-hoc analysis of performance patterns.

## C SAMPLE PROBLEM

Below, we present an example of a problem from the benchmark and discuss model performance and solution strategies from our evaluation.

**Background for reader (not included in benchmark question)** A *stable graph* is a connected graph $\widehat{\Gamma}$, multi-edges and loops allowed, together with a vertex-labeling by non-negative integers $(g_v)_{v \in V(\widehat{\Gamma})}$ satisfying that each vertex $v$ with $g_v = 0$ has valence at least 3. These combinatorial objects appear in algebraic geometry in the study of moduli spaces of stable curves, see e.g. (Schmitt and van Zelm, 2020, Section 2). The *genus* of $\widehat{\Gamma}$ is defined as $g = b_1(\widehat{\Gamma}) + \sum_{v \in V(\widehat{\Gamma})} g_v$, with $b_1$ the first Betti number (or cyclomatic number) of $\widehat{\Gamma}$.

**Question**   Given an integer $g \geq 2$, let $N_g$ be the number of isomorphism classes of stable graphs of genus $g$ with precisely 3 edges. Give a closed formula for $N_g$ valid for all $g \geq 2$.

**Solution**   To compute $N_g$, we note that each stable graph $\widehat{\Gamma}$ has an undecorated underlying graph $\Gamma$, which is one of the 10 connected multi-graphs with precisely 3 edges. Then $N_g$ can be calculated by summing over those graphs $\Gamma$ and counting the number of assignments $g_v$ to the vertices of $\Gamma$, avoiding double-counting by taking into account symmetries of $\Gamma$.

The final answer is that for $g = 2$ we have $N_2 = 2$ and for $g \geq 3$, we have

$$
N_g = \begin{cases}
\frac{1}{9}g^3 + \frac{7}{8}g^2 + \frac{5}{12}g - 2 & \text{if } g \equiv 0 \pmod 6 \\
\frac{1}{9}g^3 + \frac{7}{8}g^2 + \frac{1}{6}g - \frac{155}{72} & \text{if } g \equiv 1 \pmod 6 \\
\frac{1}{9}g^3 + \frac{7}{8}g^2 + \frac{5}{12}g - \frac{20}{9} & \text{if } g \equiv 2 \pmod 6 \\
\frac{1}{9}g^3 + \frac{7}{8}g^2 + \frac{1}{6}g - \frac{19}{8} & \text{if } g \equiv 3 \pmod 6 \\
\frac{1}{9}g^3 + \frac{7}{8}g^2 + \frac{5}{12}g - \frac{16}{9} & \text{if } g \equiv 4 \pmod 6 \\
\frac{1}{9}g^3 + \frac{7}{8}g^2 + \frac{1}{6}g - \frac{187}{72} & \text{if } g \equiv 5 \pmod 6
\end{cases}
$$

**Subquestions**   What is $N_3$? (Answer: 9) What is $N_8$? (Answer: 114) What is $N_{10000}$? (Answer: 111198615276)

**Model approaches and performance**

- GPT-5 instantly identifies the solution strategy in its first reasoning step, writing *"Essentially, I'm computing $N_g$ as a sum of connected multigraph types limited by 3 edges and considering partitions of genera"*. It performs a Python calculation to obtain the first experimental data. From theoretical considerations, it correctly identifies the shape of the final answer, writing *"Ultimately, I want a final closed formula for $N_g$ as a degree-3 quasi-polynomial with a period of 6."*. After a few attempts, it calculates this polynomial via Lagrange interpolation on datapoints with fixed residue modulo 6, discovering that the case $g = 2$ needs separate treatment. This not only represents a perfect solution to the given problem, but also mirrors precisely the approach of the human question author to solving the problem.

- GROK-4 obtains an expression for $N_g$ in a single reasoning step, though no further details are available as the GROK-4 API does not expose reasoning summaries. The model then uses a Python tool to calculate the first values and the SageMath tool to look up the resulting integer sequence in the OEIS database OEIS Foundation Inc. (2025). This being unsuccessful, it submits a very concise sketch of its answer, which is slightly less simple than the formula for $N_g$ above, as it still features a summation over $g - 2$ terms.
  In a second evaluation, GROK-4 uses the bash tool to download textbooks on algebraic graph theory and moduli spaces of curves and convert them to text. Lacking the software tools for the latter, it tries and fails to install new packages on the sandboxed Docker container, receiving an error for attempting to use sudo rights. Finally, it abandons these attempts and just submits a solution that is mostly correct, but has some small errors in one of the terms.

- CLAUDE-OPUS-4.1 also tries to combine combinatorial arguments with computer calculations in SageMath, but fails to find even the contribution from 2-vertex graphs, forgetting some topological possibilities for $\Gamma$. One noteworthy pattern is that the model includes very verbose reasoning in the form of comments and static print statements within the SageMath code.

- GEMINI-2.5-PRO starts with a correct calculation of $N_2, N_3, N_4$. However, then it makes the completely unfounded claim that *"This implies that $N_g$ is a quadratic polynomial in g."*, whereas in reality it is a *cubic quasi*-polynomial. It then submits an answer based on that wrong assumption. It does get partial credit in the subquestions for calculating $N_3 = 9$ correctly.

# D   MODEL TIERS

We evaluate models across four tiers based on their capabilities and release timeline. Tier 1 comprises current frontier models with state-of-the-art mathematical reasoning capabilities. Tier 3 includes previous-generation models that have demonstrated strong mathematical performance. Tier 4 contains legacy models included for historical comparison and baseline establishment. Currently, only models

in Tiers 1–3 are included in human grading to focus evaluation resources on the most relevant comparisons.[1]

Table 1: Models evaluated in IMProofBench, organized by tier

| Tier | Model | API Endpoint | Parameters |
|---|---|---|---|
| 1 | CLAUDE-OPUS-4.1 | `claude-opus-4-1-20250805` | `cache_prompt="auto"` `max_tokens=32000` `reasoning_tokens=31000` |
| | GPT-5 | `gpt-5` | `reasoning_effort="high"` `reasoning_summary="auto"` |
| | GEMINI-2.5-PRO | `gemini-2.5-pro` | `reasoning_tokens=32768` |
| 3 | GROK-4 | `grok-4-0709` | — |
| | O4-MINI | `o4-mini-2025-04-16` | `reasoning_effort="high"` `reasoning_summary="auto"` |
| 4 | CLAUDE SONNET 4 | `claude-sonnet-4-20250514` | `cache_prompt="auto"` `max_tokens=64000` `reasoning_tokens=63000` |
| | GPT-4O | `gpt-4o-2024-11-20` | — |
| | GEMINI-2.5-FLASH | `gemini-2.5-flash` | `reasoning_tokens=24576` |
| | GROK-3 | `grok-3` | — |
| | O3 | `o3-2025-04-16` | `reasoning_effort="high"` `reasoning_history="auto"` `reasoning_summary="auto"` `reasoning_tokens=100000` |

All models are evaluated using the Inspect framework with standardized prompting and tool access, including Python execution, web search, and SageMath for advanced mathematical computation (see App. E). The `reasoning_effort` parameter, when specified as `"high"`, enables enhanced reasoning capabilities for models that support it. The `reasoning_tokens` parameter controls the maximum length of the model's internal reasoning process, while `max_tokens` limits the total response length, including both reasoning and final answer.

# E   DETAILED TOOL DESCRIPTIONS

The evaluation environment for IMProofBench was designed to emulate the computational resources available to research mathematicians when solving complex problems. Rather than restricting models to basic arithmetic operations, we provide access to the same sophisticated mathematical software that researchers routinely use in their work. This approach reflects the reality that modern mathematical research frequently involves computational exploration, symbolic manipulation, and verification of conjectures through extensive calculation.

## E.1   TECHNICAL SPECIFICATIONS

All tools operate within the following constraints to balance computational power with practical limitations:

- **Timeout**: 15 minutes per tool invocation
- **Memory limit**: 8 GB RAM per execution
- **Environment**: Isolated Docker container running Arch Linux
- **Execution model**: Independent tool calls (no variables persist between calls), but files written to the filesystem remain accessible throughout the evaluation session

---

[1]Tier 2 is reserved for testing Command Line Interface models such as Claude Code, but implementation has been deferred to a future version of the benchmark.

## E.2   CORE COMPUTATIONAL TOOLS

### E.2.1   PYTHON ENVIRONMENT

The Python tool provides access to a comprehensive scientific computing environment (Python 3.13.7). This language was chosen for its prevalence in scientific computing and the extensive familiarity that language models demonstrate with its syntax and libraries. The environment includes standard numerical and symbolic computation packages:

- **Numerical computing**: NumPy, SciPy, pandas
- **Symbolic mathematics**: SymPy, SymEngine
- **Visualization**: Matplotlib (though output is text-based)
- **Graph theory**: NetworkX, igraph, graph-tool
- **Optimization**: CVXPY with multiple backend solvers (GLPK, ECOS, OSQP, SCS, CSDP)
- **Machine learning**: Basic scikit-learn functionality

Each Python execution runs independently with no variables or imports preserved between invocations, though files written to disk remain accessible for subsequent tool calls.

### E.2.2   BASH SHELL ACCESS

The bash tool provides command-line access to the evaluation environment, enabling models to leverage specialized mathematical software that operates through command-line interfaces. This tool serves as the gateway to domain-specific mathematical systems detailed in Section E.3.

### E.2.3   SAGEMATH

SageMath (sag, 2025) (version 10.6) serves as the primary computer algebra system, providing a unified Python-based interface to numerous mathematical software packages. Its significance in the research community stems from its comprehensive coverage of mathematical domains and its philosophy of combining the best open-source mathematics software into a coherent system.

Key features available through the `sage_computation` tool include:

- Natural mathematical syntax through automatic preparsing (e.g., `x^2` for exponentiation, `K.<a>` for field extensions)
- Extensive algebraic capabilities: polynomial rings, number fields, elliptic curves, modular forms
- Combinatorial structures: graphs, matroids, posets, designs
- Specialized packages: `admcycles` for moduli spaces of curves, `ore_algebra` for D-finite functions and recurrence operators, `pari_jupyter` for enhanced PARI/GP integration
- Integration with external systems: automatic interfacing with GAP, Maxima, PARI/GP, Singular

## E.3   SPECIALIZED MATHEMATICAL SOFTWARE

The evaluation environment includes a comprehensive suite of specialized mathematical software, accessible through the bash tool:

### E.3.1   COMPUTER ALGEBRA SYSTEMS

- **GAP** (Groups, Algorithms, Programming): Specialized system for computational discrete algebra, particularly group theory and combinatorics GAP (2024)
- **Maxima**: General-purpose computer algebra system for symbolic computation, descended from MIT's Macsyma Maxima (2025)
- **PARI/GP** (version 2.17.2): High-performance system focused on number theory computations The (2024)
- **Singular**: Specialized system for polynomial computations, commutative algebra, and algebraic geometry Decker et al. (2024)

- **Polymake** (version 4.14): System for research in polyhedral geometry and related areas Assarf et al. (2017)

### E.3.2 Algebraic and Geometric Computation

- **Normaliz**: Computation of normalizations of affine semigroups and rational cones Bruns et al.
- **LattE integrale**: Lattice point enumeration and integration over convex polytopes Baldoni et al. (2013)
- **Gfan**: Gröbner fans and tropical varieties computation
- **4ti2**: Algebraic, geometric, and combinatorial problems on linear spaces
- **msolve**: Polynomial system solving over finite fields and rational numbers

### E.3.3 Graph Theory and Combinatorics

- **nauty and Traces**: Graph automorphism and canonical labeling McKay and Piperno (2014)
- **bliss**: Another efficient graph automorphism tool
- **igraph**: Network analysis and graph algorithms library

### E.3.4 Optimization Solvers

- **Linear Programming**: GLPK (GNU Linear Programming Kit), Gurobi-compatible interfaces
- **Mixed-Integer Programming**: SCIP (Solving Constraint Integer Programs) Bolusani et al. (2024)
- **Semidefinite Programming**: CSDP, DSDP for SDP problems
- **SAT Solvers**: glucose, kissat, cryptominisat for Boolean satisfiability

### E.3.5 Proof Assistants and Verification

- **Lean** (de Moura and Ullrich, 2021): Interactive theorem prover and functional programming language
- **Mathics**: Open-source alternative to Mathematica for symbolic computation

### E.3.6 Numerical and Scientific Computing

- **Julia**: High-performance language for numerical computing
- **SciLab**: Numerical computational package similar to MATLAB
- **FLINT**: Fast Library for Number Theory
- **NTL**: High-performance number theory library

### E.4 Data Resources

The environment includes numerous mathematical databases accessible through SageMath:

- Stein-Watkins database of elliptic curves
- Jones' database of number fields
- Kohel database for elliptic curves and modular polynomials
- Cunningham tables for factorizations
- OEIS (Online Encyclopedia of Integer Sequences) integration
- Various polytope databases and mutation class data

### E.5 Web Search Capabilities

The `web_search` tool provides access to current mathematical literature and online resources. The implementation follows a provider-based architecture:

- **Internal providers**: Models from OpenAI, Anthropic, and Grok utilize their respective built-in web search capabilities, requiring no additional API keys
- **External provider**: Tavily is configured as a fallback for models without internal search capabilities (e.g., Gemini), providing AI-optimized search results

Some models, notably GROK-4, combine web search capabilities with the `wget` bash command to download full research papers for detailed analysis.

### E.6 EXAMPLE TOOL USES FROM BENCHMARK EVALUATION

Below, we list some example tool applications that occurred during our model evaluations. In each case, the full log file of the multi-turn evaluation reveals that the respective calculation played a decisive role in allowing the model to find the correct answer. To preserve benchmark privacy, we describe the relevant tool uses in general terms while leaving out the details of the specific benchmark problem.

- **Generating functions** (Model: GROK-4, Tool: SageMath)
  Solved combinatorics problem by calculating a generating function $F(x)$ and forming the exponential $G(x) = \exp(F(x))$ to extract a specific coefficient from $G$
- **Modular forms** (Model: GROK-4, Tool: SageMath)
  Compute $q$-expansion of the weight 12 cusp form $\Delta$
- **Group theory** (Model: GPT-5, Tool: GAP (2024) via Bash Shell)
  Accessed entries of the character table of a sporadic group
- **Literature access** (Model: GROK-4, Tool: Bash Shell)
  Model uses `curl` to download PDF of paper from arXiv, installs the `PyPDR2` package via `pip`, and converts the PDF to text to obtain relevant information for the benchmark problem. Note: after an initial failed attempt at installing the `PyPDR2` package, the model uses the `pip` argument `--break-system-packages` to force a user installation in the externally managed Python environment of our sandboxed evaluation environment.

## F PLANS FOR FUTURE DEVELOPMENT

Below, we give further details on our plans for the continuous development of IMProofBench.

- **Scale and outreach**: We aim to expand the benchmark to 150–300 problems, e.g. through strategic partnerships with leading mathematical institutions (e.g., MFO Oberwolfach, IAS, Fields Institute) and by recruiting domain-specific ambassadors who can promote participation at conferences and within their research networks.
- **Quality assurance and grading**: To strengthen the scientific validity of our evaluations, we will study inter-rater reliability by comparing expert gradings on the same problems. We will support graders via AI-assisted pre-screening of model answers and refine our error classification system to localize specific mistakes within solution texts rather than applying only global categories.
- **Dynamic problem management**: As mathematical knowledge evolves, problems may become easier due to new publications or techniques. We will implement a generous retirement policy allowing authors to withdraw problems affected by recent research, while regularly adding fresh problems to maintain benchmark difficulty. We also plan to release small sets of sample problems to provide the community with concrete reference points for gauging AI progress.
- **Technical innovation**: We plan to develop automated difficulty classifiers to predict which problems challenge current AI systems, explore alternative evaluation formats (such as formula reconstruction tasks and interactive problem-solving sessions), and implement bring-your-own-agent interfaces to enable companies to test internal models against the benchmark.
- **Model coverage**: Beyond proprietary frontier models, we will evaluate leading open-source reasoning systems like DeepSeek-V3.1-Terminus and Qwen3-235B-Think, promoting the strongest to Tier 1 status for human grading, ensuring long-term comparison baselines even as commercial models are deprecated. *Note*: One reason why these models were not included in this initial version of the benchmark is the ongoing challenges with enabling tool use for these models

- a requirement to put them on equal footing with other models within the inspect evaluation framework of IMProofBench.

- **Evaluation modalities**: Building on the existing IMProofBench platform and contributor network, we plan to explore further problem types and evaluation methodology. This includes:

  - combinations of informal and formalized questions and solutions (e.g., in collaboration with the ProofBench project Bowler and Carmesin),
  - specialized task formats with wide importance to mathematical research, such as formula reconstruction for sequence data of natural/rational numbers, polynomials, . . . (see e.g. Gauthier and Urban (2023); Belcak et al. (2022); D'Ascoli et al. (2022)),
  - interactive or collaborative proof attempts, including provision of hints or feedback to the model during evaluation time, more closely mimicking the setting of a researcher using commercially available AI systems.

## G    USE OF LARGE LANGUAGE MODELS

We report our use of LLMs throughout this research project. The authors take full responsibility for all content in this paper, including any LLM-assisted portions.

### G.1    WRITING AND PRESENTATION

Claude Opus was used to generate an initial draft of Sections 3 and Appendices B, D, E, G, and I and provided feedback and suggestions for our Reproducibility and Ethics Statements. Additionally, Claude Opus provided proofreading assistance and offered stylistic and structural suggestions. GPT-5 was used to collect bibliography entries of software packages in Appendix D. All LLM-generated content was thoroughly reviewed, fact-checked, and edited by the authors.

### G.2    LITERATURE DISCOVERY AND RELATED WORK

During the ideation phase and preparation of the benchmark whitepaper, we used ChatGPT o3 and Claude Opus Research to conduct comprehensive searches of the benchmarking literature and identify related projects. These tools helped surface relevant prior work and ensure thorough coverage of the existing landscape, though all citations were independently verified by the authors.

### G.3    RESEARCH IMPLEMENTATION AND DEVELOPMENT

LLMs played a substantial role in implementing the benchmark infrastructure. Claude Code, supplemented by ChatGPT's Codex CLI tool, assisted with:

- Development of the benchmark website and database architecture
- Adaptation of the Inspect framework for model evaluation
- Extraction and visualization of quantitative results

Additionally, Claude Opus provided support for organizational tasks, including meeting summaries and the creation of promotional materials. The background image for our benchmark poster was generated using GPT-5's multimodal capabilities.

All code and implementations were tested, validated, and debugged by the authors to ensure correctness and functionality.

## H    EVALUATION PROMPTS

---

**Main Question Prompt**

```
# Background
The IMProofBench project is a mathematical reasoning benchmark for AI systems, testing
their ability to solve research level math problems. Each such problem consists of one **
main question**, where the expected answer is a longform mathematical proof, and several
```

---

related **subquestions** which have short, unique answers (e.g. a natural number). The main answer will be graded by both human expert mathematicians (often the author of the question) and AI evaluators, whereas subquestion answers are checked automatically using a Python script.

# Structure of the evaluation
In the following we would like to evaluate your mathematical reasoning abilities on one such problem. The overall structure of the conversation below is that we iterate through the questions in order (main question, sub-question 1, sub-question 2, ...) and in each step, you can:

- Read the current question
- Think about it in a multi-turn environment with tool use (see below)
- Submit the answer to the current question

At each point in the conversation, you have the context of the entire previous conversation including your outputs in the thinking steps and the record of any tool uses . Note that you will *not necessarily* have access to records of your internal reasoning traces and internal tool uses, so any helpful information from these should be documented in your (external) thinking outputs.

# Multi-turn reasoning environment
To help you solve the problem, you will have access to a multi-turn conversation environment with optional tool use, based on the Inspect AI framework. At each step, you can:

- Think out loud to analyze the problem, devise a solution approach, think through the steps of mathematical arguments, etc.
- Use the `python` tool to run self-contained experiments in a standard python environment
- Use the `bash` tool to execute commands inside a docker container (running ArchLinux with some open-source mathematical software installed)
- Use the `web_search` tool to search for current information, mathematical definitions, theorems, or recent research
- Use the `sage_computation` tool for conducting an experiment in a self-contained SageMath terminal session
- Use the `submit` tool to provide your final answer to the current question (main or sub-question)

All tools have a timeout of 15 minutes, maximal memory usage (RAM) of 8 GB and run on standard 2025 hardware.

# Token constraints
You have {main_question_token_limit:,} tokens to solve the main question, and {subquestion_token_limit:,} tokens for each of the following sub-questions. This counts both your output tokens (including in tool calls) and your reasoning tokens. You are informed about your current usage after each conversation turn.

# Answer format for main question
Below you will see the text of the main question. Once you finished reasoning about it, you can register your answer using the `submit` tool. The answer for the main question should be a detailed mathematical argument, formatted in Markdown with LaTeX formulas using $...$ for inline mathematical expressions and $$...$$ for equations. Use Markdown [link formatting](https://www.markdownguide.org/basic-syntax/#links) for including online references, *not* any internal web-referencing system.

**Problem:**

{question_text}

Please work through this problem step by step. When you have your final answer, use the submit() tool to submit it.

## Subquestion Prompt

---

**Great work on the previous part!**

You have successfully completed the previous question. Now please solve the following subquestion while keeping the context of your previous work:

**Subquestion {subquestion_order}:**
{subquestion_text}

**Instructions:**
- You can reference your work from previous parts
- Use the same mathematical tools available to you

```
- When you have your final answer, use the submit() tool to submit it
- Be precise and specific in your answer format

Please proceed with solving this subquestion.
```

## Conversation Status Update

```
Please continue working on the current question. To formally register your answer, use
the 'submit' tool as per the original instructions above. Note: the conversation will
only proceed to the next stage once you use the 'submit' tool.

Token usage: {current_tokens:,} of {token_limit:,} tokens used for this stage.
```

## Python tool description

```
Use the python function to execute Python code.

The Python tool executes single-run Python scripts. Important notes:
1. Each execution is independent - no state is preserved between runs
2. You must explicitly use print() statements to see any output
3. Simply writing expressions (like in notebooks) will not display results
4. The script cannot accept interactive input during execution
5. Return statements alone won't produce visible output
6. All variables and imports are cleared between executions
7. Standard output (via print()) is the only way to see results
8. This tool has a timeout of 15 minutes and maximal memory usage (RAM) of 8 GB
```

## Bash tool description

```
Use this function to execute bash commands. Underlying system is ArchLinux with many
standard open-source computer algebra systems (like GAP) pre-installed.
This tool has a timeout of 15 minutes and maximal memory usage (RAM) of 8 GB.
```

## Web search tool description

```
Use this function to search the web for current information, mathematical definitions,
theorems, or recent research.

This tool gives you access to up-to-date information that can help with:
- Looking up mathematical definitions and theorems
- Finding recent research papers or results
- Verifying computational results against known databases
- Checking current mathematical conventions or notation
- Finding examples of similar problems or techniques

The search results will include titles, URLs, and relevant excerpts from web pages.
Use this tool when you need information that might not be in your training data or when
you want to verify facts.
```

## Sage tool description

```
Use the sage_computation function to run calculations in the open-source mathematics
software system SageMath.

The sage_computation tool executes single-run SageMath scripts. Important notes:
1. Each execution is independent - no state is preserved between runs
2. You must explicitly use print() statements to see any output
3. Simply writing expressions (like in notebooks) will not display results
4. The script cannot accept interactive input during execution
5. Return statements alone won't produce visible output
6. All variables and imports are cleared between executions
7. Standard output (via print()) is the only way to see results
8. This tool has a timeout of 15 minutes and maximal memory usage (RAM) of 8 GB

All standard SageMath functions are pre-imported and available.
The SageMath preparser is applied, so you can use natural mathematical syntax.

Key Features:
- Natural syntax: Use x^2 for powers, K.<a> for field extensions
```

```
- All mathematical objects pre-imported: Matrix, EllipticCurve, PolynomialRing, etc.
- Advanced packages available: admcycles for moduli spaces, and many more

Examples:
  # Factor a polynomial
  factor(x^100 - 1)

  # Define a number field
  K.<a> = NumberField(x^3 - 2)

  # Work with elliptic curves
  E = EllipticCurve([0, 1])
  print(E.rank())

  # Use specialized packages (example with admcycles)
  from admcycles import *
  G = StableGraph([1,1],[[1,3],[2,4]],[(1,2),(3,4)])
  print(f"Automorphisms^2: {G.automorphism_number()^2}")

IMPORTANT: Like the python() tool, you must use print() to see any output.
Nothing is returned automatically - always print your results!
```

**Submit tool description**

```
Submit your final answer for the current question or subquestion. Use Markdown + LaTeX
formatting.
The answer for the main question should be a detailed mathematical argument.

Your answer should be formatted as natural Markdown text with LaTeX formulas.
Use $ for inline math and $$ for display math, or \begin{equation} environments.
Use standard [Markdown link syntax](https://www.markdownguide.org/basic-syntax/#links)
for online references.

RECOMMENDED: Use raw strings (r''' or r"") to write LaTeX naturally without escaping.

Important formatting notes:
- Write your answer exactly as you would in a math document
- Use raw triple quotes r''' for multiline answers with LaTeX
- This lets you write \frac, \sqrt, \int naturally (no escaping needed)
- Include full mathematical reasoning with the final answer clearly stated
- Do not use custom macros (e.g., \Z, \Q, \RR, etc.). Only use valid standard LaTeX
commands
```

# I  FURTHER EXPERIMENTS AND STATISTICAL EVALUATIONS

In the section below we report on several additional evaluations, to test the effectiveness of our agent harness and the reliability of our human grading scheme, and on statistical reliability of our final-answer subquestion scores.

## I.1  ABLATION TESTS USING NON-AGENTIC EVALUATION SETUP

Our first additional experiment aims to test the effectiveness of our evaluation setup, which uses the inspect framework and with tool access as described in App. E. For this we conducted ablation tests where models were presented with the main question, followed sequentially with all the subquestions, but with

- minimal additional prompting, and single-turn thinking (answer is given by single model reply),

- either no tools or selected sets of their native tools (like web search and code interpreter) hosted by the API provider.

More precisely, the initial message to the model is simply the main question text. After an answer is received, the follow-up prompt introducing the first subquestion is:

**Non-agentic prompt for first subquestion**

```
Thanks! Below I'll ask some follow-up questions, where answers will be parsed
automatically. They might repeat the main question or ask for results in special cases.
Please provide the final answer in a \boxed{} environment for easier extraction. This
desired answer will typically be short, either \boxed{Yes} or \boxed{No} or a numerical
result like \boxed{172} or \boxed{-13/45}. Do not use additional (LaTeX) formatting
within the \boxed environment.

Subquestion 1:
{subquestion_text}
```

Any further subquestions are presented as follows:

**Non-agentic prompt for further subquestions**

```
Thanks!
Subquestion {number}:
{subquestion_text}
```

In Table 2 we compare the performance of GPT-5 inside inspect (left) with the ablation setup given various tools (right). We see that the best performance is actually achieved by only granting web

|         |            | Code Interpreter | |
| ------- | ---------- | ---- | ---- |
| Inspect | Web Search | No | Yes |
| 52.5%   | No         | 55.0% | 47.6% |
|         | Yes        | 57.4% | 51.2% |

Table 2: Average final-answer subquestion scores

search, surpassing the score inside the inspect setup. In fact a striking pattern is that in the $2 \times 2$ grid on the right, granting web search leads to a boost of $2 - 3$ percentage points, whereas granting code interpreter access actually *decreases* model performance by $6 - 9$ percentage points.

As a second experiment we compared the inspect setup with a no-tools ablation: While the score

| Model         | Inspect | No Tools |
| ------------- | ------- | -------- |
| GEMINI-2.5-PRO | 38.1%  | 39.4%    |
| GROK-4        | 60.8%   | 44.4%    |

Table 3: Inspect vs. no-tool ablation results on final-answer subquestions

of GEMINI-2.5-PRO is basically unchanged, the example of GROK-4 shows that the inspect agent framework can significantly boost the performance of some models.

### I.2 PRELIMINARY ANALYSIS OF HUMAN GRADING RELIABILITY

To assess inter-rater reliability of our human grading scheme, we conducted a preliminary analysis during which 15 questions were independently graded by multiple evaluators (11 questions by 2 graders, 4 questions by 3 graders), yielding 55 pairwise answer comparisons. Each grader worked blindly, with model identities hidden behind randomized aliases.

For the overall progress grade (scored 0–3), graders achieved exact agreement in 50.9% of cases and agreed within one point in 89.1% of cases, suggesting that the coarse-grained progress assessment is reasonably reliable. At the question level, 18.2% of questions showed high overall agreement ($\geq 80\%$), 72.7% showed moderate agreement (60–79%), and only 9.1% showed low agreement ($< 60\%$).

Agreement varied considerably across the eight binary grading categories, as shown in Table 4. Objective categories such as "Correct End Result" showed high agreement (84.9%), while more subjective error classifications like "Incorrect Logic" and assessments of "Mathematical Insight" proved harder to pin down (both 58.2%). This suggests that while graders largely agree on whether a solution succeeds and makes meaningful progress, diagnosing the precise nature of errors in failed attempts involves more subjective judgment.

| Category | Agreement |
|---|---|
| Correct End Result | 84.9% |
| Hallucinated Facts | 74.5% |
| Problem Understanding | 72.7% |
| Useful Progress | 70.9% |
| Calculation Error | 65.5% |
| Conceptual Error | 61.8% |
| Mathematical Insight | 58.2% |
| Incorrect Logic | 58.2% |

Table 4: Inter-rater agreement by grading category, based on 55 pairwise comparisons across 15 questions.

### I.3 STATISTICAL UNCERTAINTY FROM STOCHASTIC LLM BEHAVIOR

For any AI benchmark, the reported model scores are empirical estimates subject to statistical uncertainty. Following Miller (2024), there are two distinct sources of this uncertainty:

1. **Finite question sampling:** The benchmark questions represent a finite sample from a hypothetical super-population of all questions testing the relevant skills. Even with deterministic model behavior, the score on $n$ questions would differ from the "true" score on that super-population.

2. **Stochastic model behavior:** Due to temperature sampling and other sources of randomness, a model may produce different answers—and hence different scores—when presented with the same question multiple times.

The framework of Miller (2024) addresses the first source by computing standard errors from the variance of scores between the actual set of test question and the whole hypothetical super-population. However, this approach has limitations for our purposes: the resulting confidence intervals depend only on the mean score and number of questions, adding no information beyond what is already visible in the results. Moreover, such intervals could easily be misinterpreted as quantifying measurement noise from stochastic model behavior, when they actually capture uncertainty about the question sample.

We therefore focus on directly measuring the second source of uncertainty. For our Tier 1 models, we conducted two independent evaluations per question, allowing us to estimate the variance attributable to non-deterministic LLM behavior. Specifically, for each model $M$ and question $Q$, we compute the sample variance $V(M, Q)$ of the weighted subquestion scores across the two evaluations. We then estimate the standard deviation of the average score as

$$\sigma(M) = \frac{1}{N} \sqrt{\sum_Q V(M, Q)},$$

where $N$ is the number of questions. The results are shown in Table 5.

The standard deviations range from approximately 3 to 4 percentage points across models, indicating that the stochastic nature of model responses contributes meaningful but moderate uncertainty to the final scores. A 95% confidence interval for each model's score can be approximated as $\pm 2\sigma$, i.e., roughly $\pm 6$–$8$ percentage points.

| Model | Score | Std. Dev. ($\sigma$) |
|---|---|---|
| GROK-4 | 60.8% | 3.54 |
| GPT-5 | 53.6% | 3.8 |
| GEMINI-2.5-PRO | 38.1% | 4.1 |
| CLAUDE-OPUS-4.1 | 36.4% | 3.1 |

Table 5: Estimated standard deviations of average subquestion scores due to stochastic LLM behavior, based on two evaluations per question (for $n = 44$ questions with at least one subquestion).

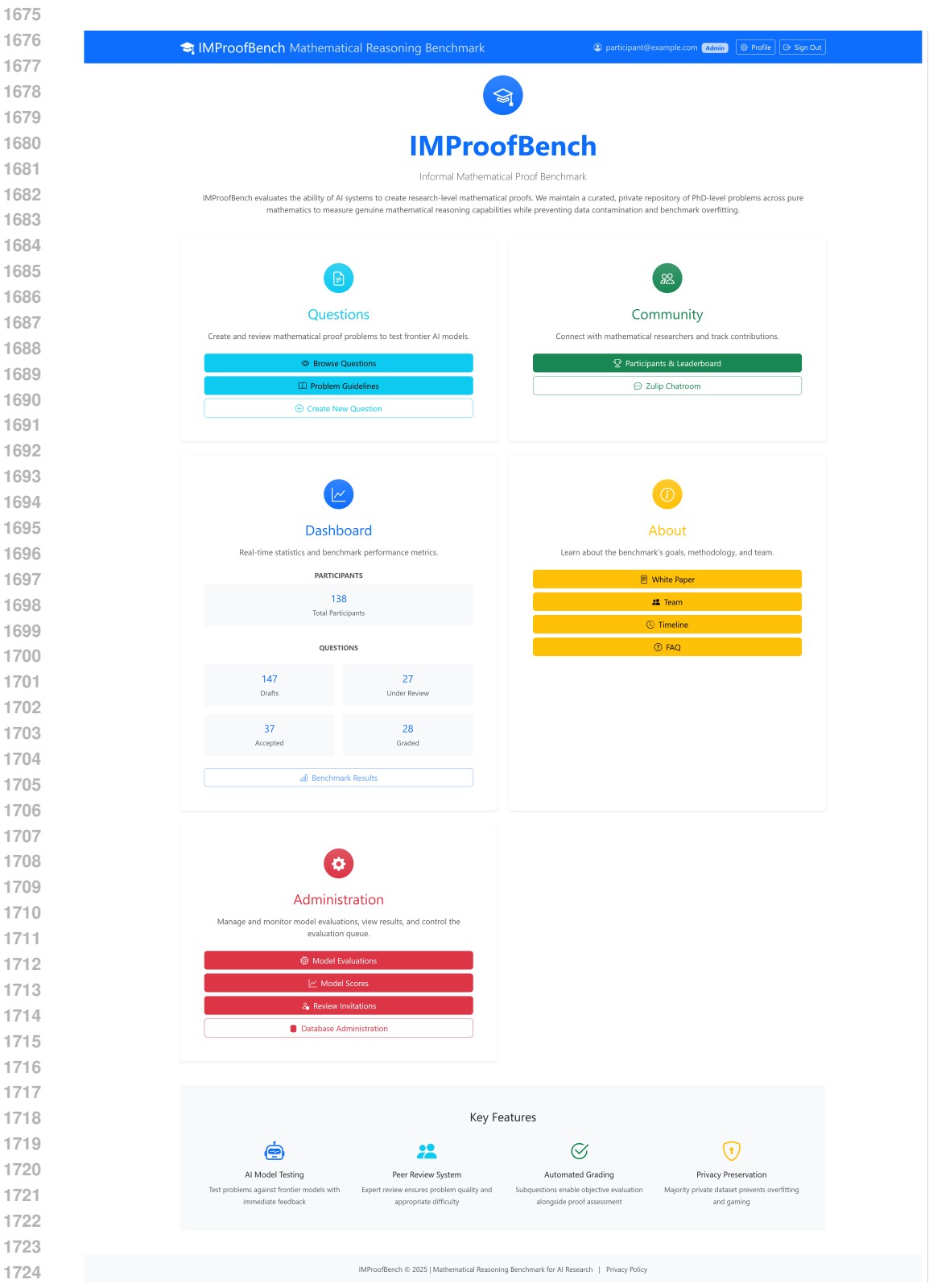

Figure 14: Landing and overview page of IMProofBench website.

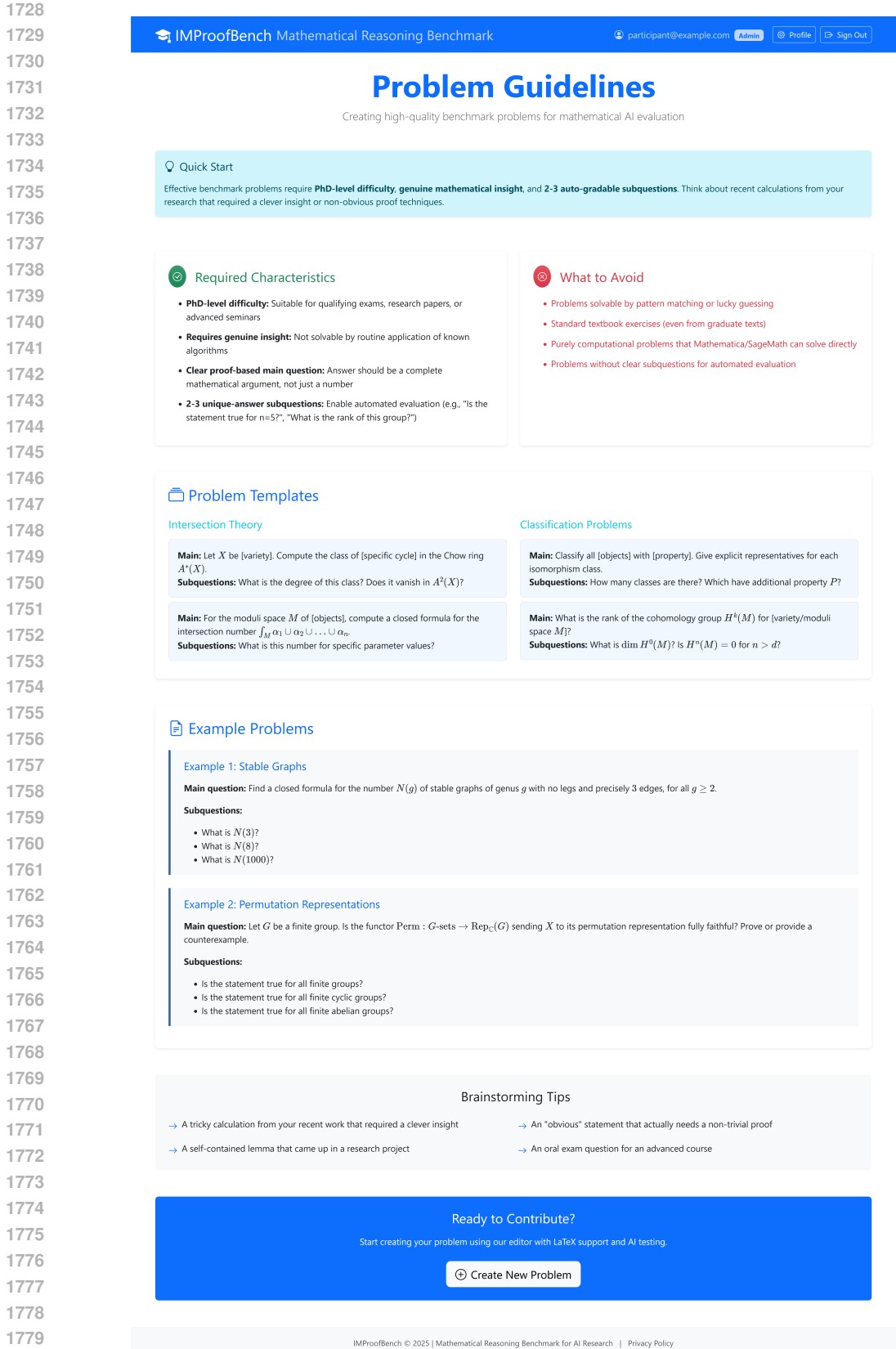

Figure 15: Guidelines for authoring benchmark problems.

Figure 16: Window for editing questions, solutions, and their associated subquestions; via the blue button, the user can request up to 20 free AI solution previews per day to check the suitability of the question.

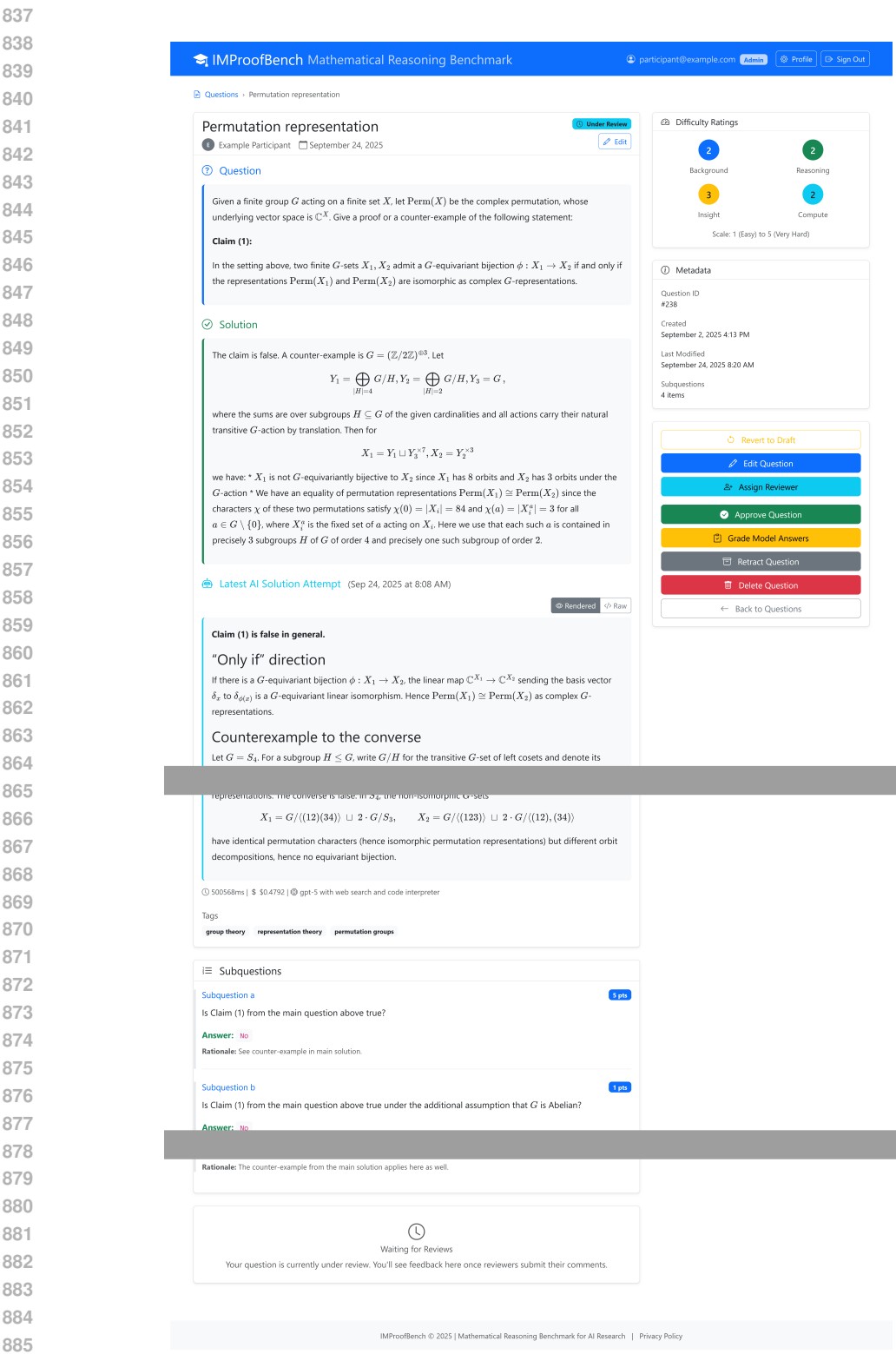

Figure 17: Overview page of question data (with main question, sample solution, AI answer preview, and subquestions).

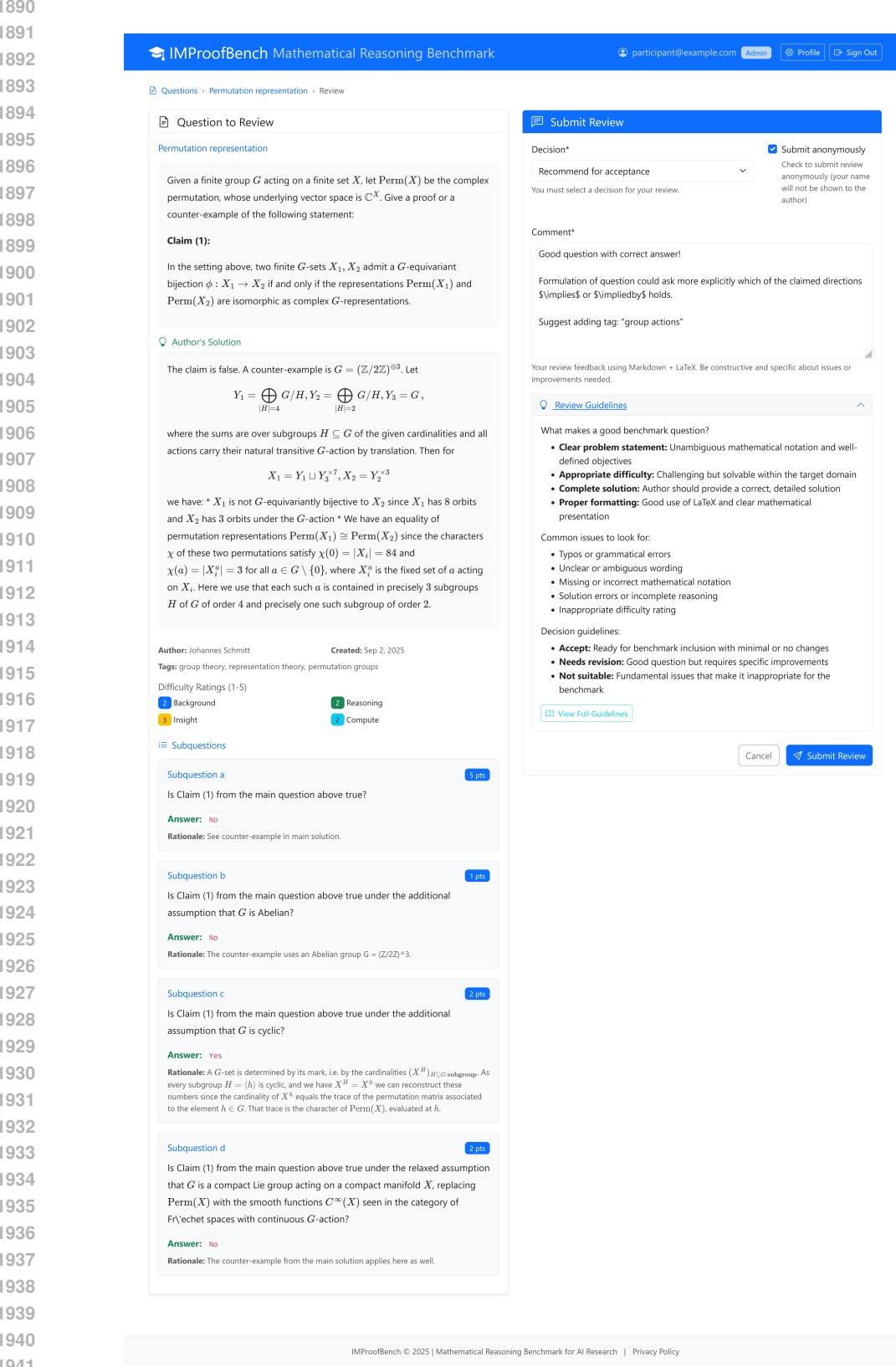

Figure 18: Question review window showing text box for feedback and review instruction summary.

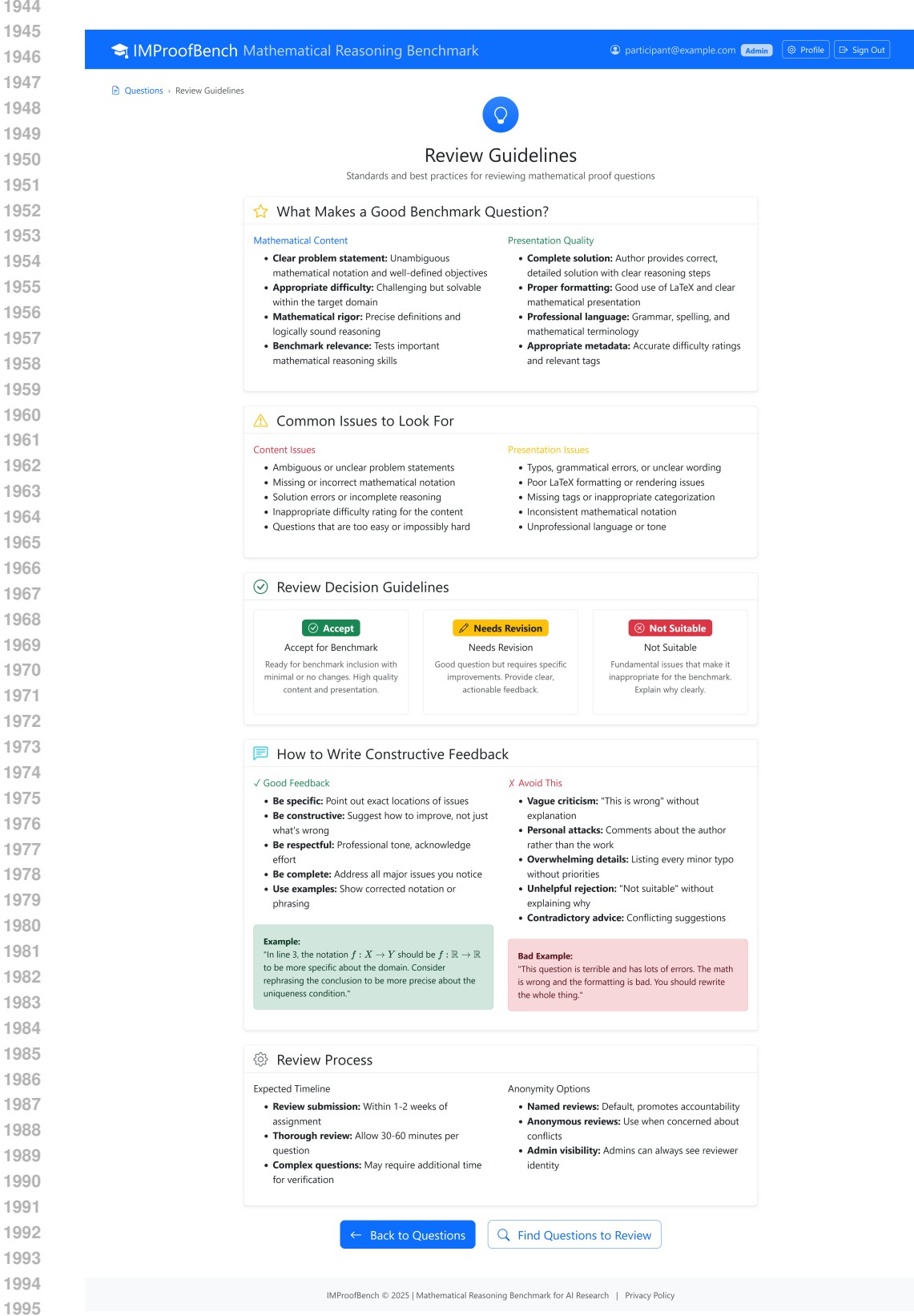

Figure 19: Detailed explainer of review instructions and process.

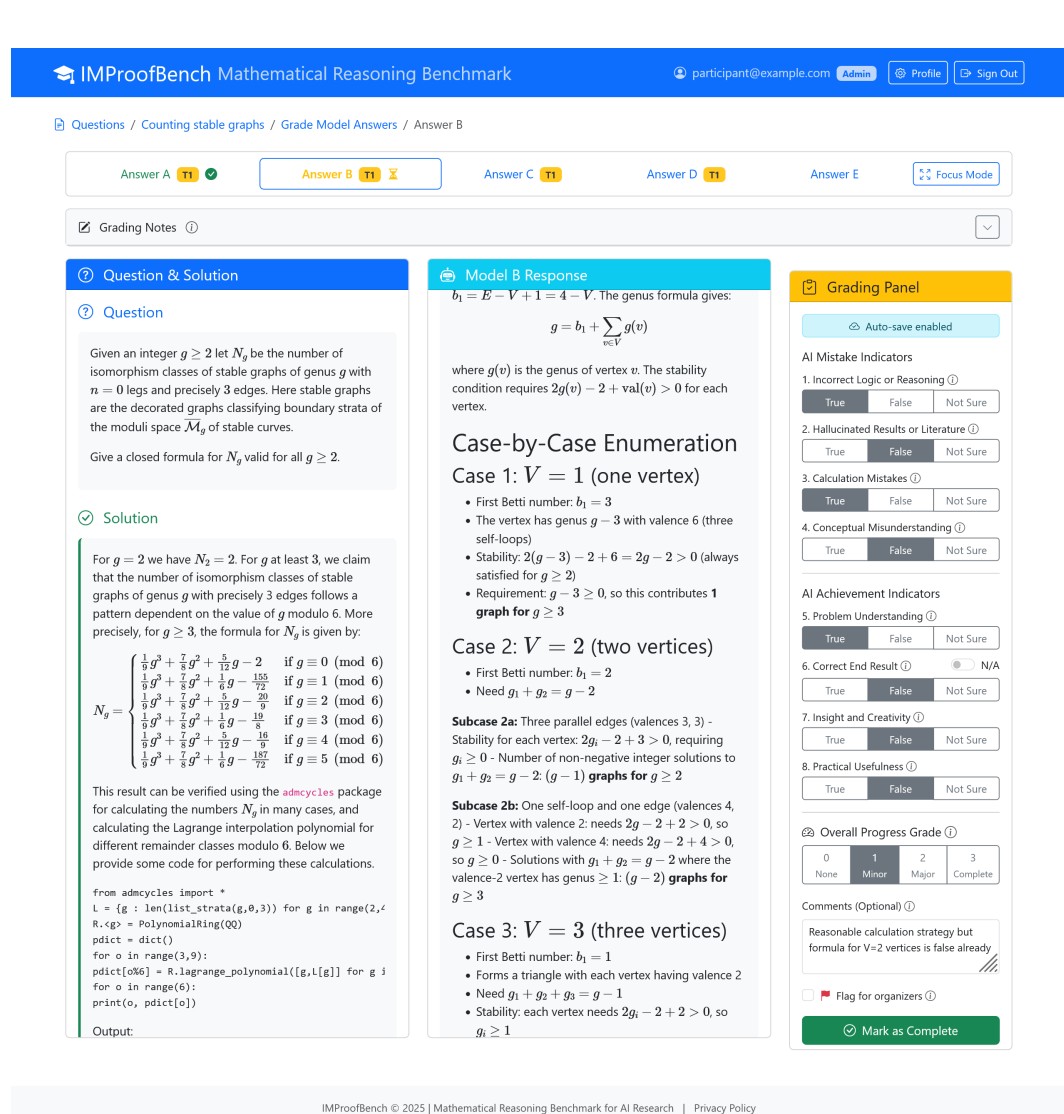

Figure 20: Grading form, displaying sample solution, model answer, and scoring form side by side. Model identities (A–E) at the top are randomized on starting the grading, and only revealed when grading is complete, to avoid bias.

