# OpenReview forum: "IMProofBench: Benchmarking AI on Research-Level Mathematical Proof Generation"
_ICLR.cc/2026/Conference — Submitted to ICLR 2026_

### Official Review · Reviewer_H7Lc · 2025-10-15

**Soundness:** 3
**Presentation:** 4
**Contribution:** 4
**Rating:** 6
**Confidence:** 5

**Summary:**

The paper introduces IMProofBench, a benchmark for evaluating LLMs on research-level mathematical proof generation rather than only final answers. The current release contains 39 peer-reviewed problems contributed and reviewed by professional mathematicians, each paired with auto-gradable follow-up subquestions. Models are evaluated in an agentic framework with access to tools (Python, Bash/GAP/Maxima, SageMath, and web search) and generous token budgets; main proofs are graded by the problem author on a 0–3 scale, while follow-ups are auto-graded. On this benchmark, GPT-5 achieves complete, fully correct proofs on 22% of tasks; Grok-4 attains the best final-answer accuracy (52%). There are also some qualitative and quantitative analyses showing that the final answer evaluations are informative but insufficient, LLMs are still struggling with open problems, etc.

**Strengths:**

- **Clear problem framing & High-fidelity evaluation setup:** Directly targets proof writing at the research level, filling a documented gap left by final-answer benchmarks and high-school/undergrad datasets. Agentic environment with real tools (SageMath, Python, web search), submit tool to separate reasoning from final answers, and substantial token budgets appropriate for deep math. This successfully simulates the real mathematician's working scenario.
- **Quality control & community process:** Problems are authored and peer-reviewed by domain experts; the site/workflow formalizes submission, review, and grading instructions.
- **Forward-looking & contamination-aware:** Authors are considering dynamic retirement of problems to maintain fair evaluations. Human mathematicians in the loop could make the LLM evaluation promising and truly tailored to the needs of human users.

**Weaknesses:**

- **Tool-access parity and fairness:** Different models use different web-search providers (internal vs external) and may leverage shell utilities to fetch PDFs, raising parity questions about retrieval power across vendors
- **Potential grader bias & reliability issues:** Main proofs are graded by the problem’s author; without inter-rater studies, there’s risk of variability or unconscious bias.

**Questions:**

- **Contamination controls:** For the dynamic problem management system, how will you detect that a problem has become easier due to new publications, and how often will items be rotated or retired?
- **Open-problem protocol:** Since no open problems were solved here, will future releases track partial advances on open questions differently (e.g., author judgment of novelty), and how will you guard against over-claiming?
- **Scaling human effort:** As the benchmark grows, how will you mitigate grader load while preserving rigor (e.g., rubric refinements, rubric-guided LLM pre-screening, partial formalization)?

---

> ### Author Response · Authors · 2025-11-21
>
> We thank the reviewer for their feedback. We are happy to hear they appreciate our high-fidelity setup, our quality control and community process, and our forward-looking and contamination-aware procedure. Below, we address each of the reviewer’s questions and remarks in detail.
>
> **How can we ensure tool-access parity since different models use different web-search providers?**
>
> Models are often finetuned to work better with the web-search providers or web-search tools that are used internally by the model provider. Therefore, we consider the entire API implementation as the model, as we wish to compare performance from a user’s perspective: a user will also use that model with the built-in tools. If these tools are better implemented by the model provider, then this can provide a significant advantage for their model, which we think is an important consideration that is part of the process.
>
> **Can you perform consistency checks across graders to ensure reliability?**
>
> We are right now in the process of performing a second grading round: we reached out to our question reviewers asking them to conduct an independent grading on their reviewed questions, and the mathematicians among the core team members will also perform independent gradings on a selection of questions whose formulation and solution is accessible to all of them (for very specialized questions, only the authors, reviewers and other select experts are expected to be qualified to grade the model answers).
> To give the reviewers some time to perform this task, we will report the results in the final revision of the submitted paper (planned for November 28th).
>
>
> **How will you detect that a problem has become easier due to new publications and how often will items be rotated or retired?**
>
> Thanks, this is an excellent point and we are happy to elaborate. The main mechanismto detect problem difficulty changes are the planned future grading sessions. These will occur every 4-8 months depending on the schedule of new model releases, to balance the time commitments of our graders with our plan to provide timely updates on novel AI models. In the invitations to these emails, we ask the question authors to take a moment to review the question and verify that its difficulty has not been compromised by new publications. We also give them the opportunity to flag the question for retirement e.g. due to own work that they have published or intend to publish imminently. Beyond the author’s own assessment based on their knowledge of the literature, we also expect that the proof answers of frontier models would contain references to recently published papers if those papers reduced the question difficulty (by providing a shortcut to the intended solution path).
>
> **How will you keep rigorous track of partial advances on open problems?**
>
> Since the problems are open, even the problem author does not know the full correct answer, and it is therefore hard to differentiate between various approaches that might not be entirely correct, but could potentially lead to a solution. The problem author can however use the minor progress or major progress categories if the model books a small amount of progress, or even so much progress that the problem author can now solve it, respectively.
>
> **How will you mitigate grader load while preserving rigor?**
>
> We fully agree that the amount of work involved in grading any individual model answer would make it challenging or infeasible to perform the fine-grained proof-based evaluation of  every single newly released model. However, both the initial design of the benchmark, as well as some of the planned future developments are shaped to mitigate this limitation:
> * Final-answer evaluation of subquestions can serve as a good proxy for model performance on the main question (correlation of 0.45, see also below) and allows an easy-to-evaluate method to obtain a relative ranking of the tested models. For the continuing development of the benchmark, the selection of further models for human grading will be guided by this measure. This ensures that the grading effort focuses on models close to the frontier for mathematical reasoning, which we consider to be the most interesting.
> * As for grader motivation, we have observed very high turnouts: at the time of writing, 45 of the 48 accepted questions have been graded by their authors. This can be explained by the intrinsic interest of the graders in the questions which they designed. On the other hand, the work is distributed on many shoulders, with the 48 questions having 30 authors. Apart from the intrinsic diversity this provides, it also means that the total amount of work per person stays very manageable.
> * As demonstrated in recent work [1,2], LLMs have the potential to very reliably grade longform mathematical proofs. In future work, we plan to use our human grader data to develop, calibrate and test such automated grading systems that would allow more scalable evaluations.

---

### Official Review · Reviewer_Bjxz · 2025-10-30

**Soundness:** 2
**Presentation:** 3
**Contribution:** 2
**Rating:** 6
**Confidence:** 3

**Summary:**

IMProofBench contains 39 problems authored and reviewed by expert mathematicians; each main question asks for a full proof and comes with FA subquestions for scalable scoring. Evaluation occurs in an agentic framework with Python, Bash, SageMath, and web search; models submit a final answer distinct from intermediate reasoning; token budgets are 300k for the main question and 100k per follow-up. Reported headline numbers: GPT-5 achieves 22% complete solutions on proof tasks; GROK-4 achieves 52% FA accuracy on the subset with follow-ups; the correlation between human proof scores and FA scores is 0.45.

The paper positions itself against prior math benchmarks focused on high-school/undergrad or FA-only tasks, and argues for proof-based, research-level evaluation with human graders. Core contributions are explicitly listed as the benchmark itself, a cross-model proof study, and a qualitative analysis of errors and strengths.

**Strengths:**

- Clear gap and strong motivation. The need to evaluate research-level proof writing rather than only FA questions is well motivated in the abstract and introduction. The benchmark directly targets that gap and brings expert graders into the loop.

- Realistic evaluation environment. The agentic setup—with Python, SageMath, Bash, and web search—matches how a human would work. The required “submit” step separates scratch work from the final proof, which helps graders. The token budgets are large enough to explore tools without being trivially restrictive.

- Thoughtful human grading design. Authors grade main proofs on a 0–3 progress scale and also tag error types (logic, concept, hallucination, calculation) and achievement indicators (understanding, insight, usefulness). This captures more signal than a single pass/fail.

- Quantitative and qualitative results. The paper reports both proof-score distributions and FA accuracy, plus the 0.45 correlation between FA and human proof scores, making a concrete case that FA alone is an imperfect proxy. The qualitative section usefully documents typical failure modes (confident but incorrect claims, “well-known result” shortcuts) and behavior (rare abstention).

- Problem sourcing and review. Problems go through an authoring pipeline with reviewer feedback and acceptance only after concerns are resolved; guidelines stress PhD-level insight and proof-centric tasks, with FA subquestions for automated scoring.

**Weaknesses:**

- Private and evolving dataset. The benchmark is private, with plans to open-source the platform code and release open sample problems; for ICLR, private data can limit reproducibility and independent verification. Because the set is evolving, comparability across time may drift unless a frozen v1.0 test split is committed.

- Scale and composition. N=39 is small for a general benchmark, and the paper itself highlights this. The topic distribution currently leans toward areas favored by organizers. Both factors raise questions about coverage and selection bias relative to “research-level mathematics” as a whole.

- Human-in-the-loop reliability. The protocol records rich grader signals, but the paper does not report inter-rater reliability or consistency checks across graders for the same solution. Without IRR, it is difficult to quantify grading variance and potential systematic bias.

- Agent and tool confounds. Allowing web search and shell access reflects reality, but it also introduces test-time retrieval confounds (e.g., locating key lemmas on the web). The appendix even notes models downloading papers via wget/curl. This is realistic but complicates claims about reasoning versus retrieval. A controlled ablation (no web; no Bash; SageMath only; different token budgets) would clarify how much performance comes from tool-mediated lookup versus algebraic reasoning.

- Evaluation fairness across model APIs. The paper mentions that some models (e.g., GROK-4) give short final answers and have hidden reasoning tokens, complicating assessment. Tiering and API differences can affect both behavior and resource use. More detail on API normalization and prompting parity would strengthen fairness claims.

- Statistical characterization. The headline metric “% complete solutions” on 39 items is informative but noisy. Confidence intervals, per-area breakdowns, and problem-level difficulty calibration would help. The 0.45 correlation between FA and proof scores is useful; expanding this analysis (e.g., partial correlations by topic, difficulty, or tool usage) would improve interpretability.

- Reproducibility timing. The paper commits to releasing platform code and open sample problems before Nov 30, 2025, which would ease review concerns if delivered, but program committees must judge the submission as it stands. A clear plan for a frozen public subset and exact grader guidelines would materially improve reproducibility.

**Questions:**

See Weakness

---

> ### Author Response · Authors · 2025-11-21
>
> We thank the reviewer for their feedback. We are happy to hear they find we address a clear gap, value our realistic evaluation environment with a good human grading design, and our analysis of both quantitative and qualitative results. Below, we address each of the reviewer’s questions and remarks in detail.
>
> **Why did you opt for private benchmarking?**
>
> See Q1 of our main reply.
>
> **How does the small scale affect your analysis?**
>
> See Q2 of our main reply.
>
> **How does the current topical skew towards algebraic geometry affect your analysis?**
>
> See Q3 of our main reply.
>
> **Can you perform consistency checks across graders to ensure reliability?**
>
> We are right now in the process of performing a second grading round: we reached out to our question reviewers asking them to conduct an independent grading on their reviewed questions, and the mathematicians among the core team members will also perform independent gradings on a selection of questions whose formulation and solution is accessible to all of them (for very specialized questions, only the authors, reviewers and other select experts are expected to be qualified to grade the model answers).
> To give the reviewers some time to perform this task, we will report the results in the final revision of the submitted paper (planned for November 28th).
>
> **Can you perform an ablation to compare the current agentic setup with a model without any tools?**
>
> Thank you for this excellent suggestion! We ran such tests with models
> * GPT-5 (no tools/web search only/code interpreter only/search and code interpreter)
> * GPT-5 Pro (web search)
> * Gemini 2.5 Pro, Grok 4, DeepSeek V3.1 Terminus (no tools)
> The results on weighted subquestion scores show an interesting pattern:
> * GPT-5 (no tools/web search) fall within 2 percentage points of our GPT-5 inspect agent, which lies within the expected 1-sigma interval (3.8 percentage points) from variability of answers
> * GPT-5 (code interpreter/search+code interpreter) fall *below* that interval (-7.3 and -5.1 percentage points, respectively), indicating that the code interpreter actively harms the performance
> * GPT-5 Pro and Gemini 2.5 Pro without tools only differ by 0.3 percentage points from their inspect agent performance, indicating that the model cannot effectively make use of this agent environment
> * Conversely, the two remaining models exhibit large performance drops from removing the inspect framework: Grok 4 (58.5% to 42.2%) and DeepSeek V3.1 Terminus (35.7% to 31.3%). This demonstrates that some models (and in particular our leading model Grok 4 on subquestion performance) profit substantially from the tools provided by the inspect framework we set up.
>
> A detailed discussion will follow in the revision of our paper, planned for Nov 28th.
>
>
> **Can you perform statistical analysis on your results?**
>
> Thank you for pointing this out! We will add error bars to the evaluations of our Tier 1 models (GPT-5, Grok 4, Gemini 2.5 Pro, Claude Opus 4.1) that capture the variance in average subquestion scores coming from the two independent evaluations we conducted for these models. This type of error captures the variability of evaluation results coming from the stochastic nature of LLM outputs. Final plots will appear in the Nov 28th revision, but the standard deviations range from 3.4 to 4 percentage points, with average model scores ranging from 33% to 58%. Thus despite the modest benchmark size, the statistical resolution is already quite good.
>
> For other models, we only performed one evaluation per (model, question) pair, so that no such variability data is available.
>
> **Can you elaborate on the correlation between final answer and proof scores?**
>
> Unfortunately, our current dataset is not large enough to provide sufficiently significant results on the correlation between final answer and proof scores when broken down by categories.
>
> **Can you provide a clear frozen public test set?**
>
> To increase transparency and accountability, we plan to:
> Create a publicly accessible archive of monthly snapshots summarizing the statistics of IMProofBench (all model scores listed in this paper, correlations, etc) and make it available on the IMProofBench website
> When open-sourcing the repository on November 30th, the repository will contain a sample of 5 representative example problems together with solutions, which allows external parties to verify our evaluation infrastructure. We also commit to updating this public sample set to grow in size so it represents at least 10% of the questions accepted in the benchmark.
>
>
> **Can you provide the exact grader guidelines?**
>
> Yes, Appendix B along with the screenshots included at the end of the paper specify the full grader guidelines.

---

### Official Review · Reviewer_YB3b · 2025-11-01

**Soundness:** 3
**Presentation:** 2
**Contribution:** 2
**Rating:** 4
**Confidence:** 3

**Summary:**

The paper presents **IMProofBench**, a private, evolving benchmark aimed at **research‑level, proof‑centric** evaluation of LLMs in an agentic environment with tools (Python, SageMath, Bash, web search) and human expert grading on a 0–3 rubric (peer‑reviewed problem pipeline; automated grading for follow‑up subquestions). Key results: among 39 problems (with 79 subquestions), GPT‑5 achieves 22% complete proofs, while GROK‑4 leads on final‑answer subquestions with 52% perfect accuracy (Figs. 5 & 4; §§3–4; App. E; pp. 4–9, 21–23). Overall, the work targets an under‑served evaluation axis—proof quality rather than answer‑only scoring (motivation and setup; §1–§3; Fig. 3), but it currently has limited scale, private data, and heterogeneous tool/search access that may confound cross‑model comparisons (State & limits; §§3.3–3.4; App. A/E; pp. 5–9, 14–24).

**Strengths:**

1. **Proof‑centric, research‑level focus**

Articulates a gap in answer‑only benchmarks and centers proof writing with human expert grading (Intro; §1; §3.3), clarifying capabilities beyond final answers (novel evaluation dimension). Problems span pure/applied topics; 7 are flagged “open,” signaling frontier intent (State; §3.4; p. 6)—useful for probing limits.   - Worked sample (stable graphs) illustrates target rigor and expected solutions (App. C; pp. 19–20), aiding clarity and reproducibility expectations.

2. **Documented pipeline and blind expert grading**

Two‑stage review with admin + domain expert; acceptance only after “no further comments” (Fig. 2; §3.2; pp. 4–5) → quality control.

3. **Balanced quantitative and qualitative evaluation**

Reports proof vs final‑answer outcomes and a 0.45 correlation between author‑weighted subquestion score and 0–3 progress (§4.1; Figs. 4–5; p. 7) → answer accuracy ≠ proof quality.

**Weaknesses:**

1. **Private dataset limits replication and audit**
- Dataset is private; only code + sample problems planned for release (Reproducibility; p. 10) → third‑party re‑grading is constrained.
- No public leaderboard or downloadable test set is provided (No direct evidence found in the manuscript) → weakens community tracking.
- Human‑graded proofs cannot be independently re‑assessed at scale without access to items/outputs (§3.3; p. 5–6; Reproducibility; p. 10).
2. **Small scale and topical skew**
- 39 problems from 23 contributors is modest for statistical comparisons (§3.4; p. 6) → low power.
- Tag distribution dominated by algebraic geometry (App. A Fig. 12; pp. 14–15) → representativeness concerns.
- Only 7 “open” problems (§3.4; p. 6), so “research‑level” spans a broad difficulty band → interpretability issues.
3. **Relation to similar benchmarks is not fully disentangled**
- FrontierMath also targets advanced/research‑level problems but emphasizes final‑answer evaluation; fairness concerns around access are documented (Related Work §2; arXiv; EpochAI clarification). The paper cites FrontierMath (§2) but lacks side‑by‑side comparisons or controlled discussion beyond final‑vs‑proof framing (No direct evidence of direct cross‑eval in manuscript).

**Questions:**

Can you situate IMProofBench against FrontierMath/RealMath/HLE/UQ/USAMO/PutnamBench/MiniF2F with a small cross‑evaluation or calibrated narrative to clarify novelty and complementarity? (Related Work §2).

---

> ### Author Response · Authors · 2025-11-21
>
> We thank the reviewer for their feedback. We are happy to hear they appreciate our proof-centric, research-level focus, our pipeline with blind expert grading, and our analysis containing both quantitative and qualitative evaluation. Below, we address each of the reviewer’s questions and remarks in detail.
>
> **Why did you opt for private benchmarking?**
>
> See Q1 of our main reply.
>
> **How does the small scale affect your analysis?**
>
> See Q2 of our main reply.
>
> **How does the current topical skew towards algebraic geometry affect your analysis?**
>
> See Q3 of our main reply.
>
> **Can you clarify how your work complements prior work, in particular with respect to FrontierMath?**
>
> As discussed in our related work, we mainly differ from FrontierMath in two significant areas:
> * We evaluate proof-based reasoning and not just final-answer accuracy. Prior work [1,2,3,4] has consistently shown that final-answer problems are not a great indicator for proof-based reasoning (although they do give some indication). Our work further confirms this for research-level mathematics: the correlation between proof-based performance and final-answer accuracy is 0.45, which is clearly positive, but also indicates that final-answer accuracy does not tell the entire story.
> * FrontierMath is supported by OpenAI, and they share most problems and answers with the company. This led to various questionable practices (see, e.g., [1]). Therefore, they are not independent. While we do seek out contributions by companies, we do not (and never will) share questions for this purpose, and we will always remain independent from any financial contributions we receive.
>
> [1] https://siliconreckoner.substack.com/p/the-frontier-math-scandal

---

### Official Review · Reviewer_xDmc · 2025-11-01

**Soundness:** 2
**Presentation:** 2
**Contribution:** 2
**Rating:** 4
**Confidence:** 3

**Summary:**

This paper introduces IMProofBench, a private benchmark consisting of 39 peer-reviewed problems developed by expert mathematicians, designed to assess LLMs on research-level mathematical proof writing rather than just final-answer prediction. Each problem requires a detailed proof and is paired with subproblems that have final answers. The authors evaluate LLMs on IMProofBench in an agentic environment, giving models access to Python, SageMath, and web tools for reasoning and computation. Human mathematicians grade main proof tasks, while subproblems are auto-scored. Experiments show that GPT-5 achieves the best overall performance, solving 22% of full proofs, while GROK-4 leads in final-answer accuracy. Analysis reveals that LLMs are prone to reasoning errors, ranging from simple logical mistakes to deep misconceptions, and they frequently hallucinate existing results to obtain a flawed answer.

**Strengths:**

1. The scope of the IMProofBench is novel, aiming to measure LLM's capability in research-level proof generation rather than just problem-solving accuracy.
2. The problems peer-reviewed by mathematicians in IMProofBench would be a valuable resource for the LLM research community
3. The evaluation is done in an agentic setting rather than pure LLM reasoning, reflecting a more realistic research workflow.
4. The authors provide valuable insights into current frontier LLMs on research-level proof generation, reveals that LLMs are prone to reasoning errors, ranging from simple logical mistakes to deep misconceptions, and they frequently hallucinate existing results to obtain a flawed answer.

**Weaknesses:**

1. The scale of IMProofBench is extremely small, containing only 39 problems.
2. The reliance on experts for proof evaluation limits its scalability and usefulness for the research community in model evaluation and development.
3. While the authors evaluate frontier LLMs in an agentic setting, it would be nice to compare their performance in a single-model reasoning mode without tools, to ablate the effect of tool use in the agentic setup.
4. Each problem in IMProofBench is paired with subproblems that have final answers that can be auto-graded. What is the correlation between subproblem correctness and the correctness of the entire proof? Could this serve as a proxy for overall proof correctness when lacking human expert graders?

**Questions:**

1. While the authors mention community outreach to recruit professional mathematicians in the paper, what are their qualifications?
2. Is there plan to host a leaderboard for IMProofBench? If so, do you need professional mathematicians to grade every submission?

---

> ### Author Response · Authors · 2025-11-21
>
> We thank the reviewer for their feedback. We are happy to hear they find our scope novel, our peer-reviewed problems a valuable resource, and that we provide valuable insights. Below, we address each of the reviewer’s questions and remarks in detail.
>
> **How does the small scale affect your analysis?**
> See Q2 of our main reply.
>
> **Can you perform an ablation to compare the current agentic setup with a model without any tools?**
> Thank you for this excellent suggestion! We ran such tests with models
> * GPT-5 (no tools/web search only/code interpreter only/search and code interpreter)
> * GPT-5 Pro (web search)
> * Gemini 2.5 Pro, Grok 4, DeepSeek V3.1 Terminus (no tools)
> The results on weighted subquestion scores show an interesting pattern:
> * GPT-5 (no tools/web search) fall within 2 percentage points of our GPT-5 inspect agent, which lies within the expected 1-sigma interval (3.8 percentage points) from variability of answers
> * GPT-5 (code interpreter/search+code interpreter) fall *below* that interval (-7.3 and -5.1 percentage points, respectively), indicating that the code interpreter actively harms the performance
> * GPT-5 Pro and Gemini 2.5 Pro without tools only differ by 0.3 percentage points from their inspect agent performance, indicating that the model cannot effectively make use of this agent environment
> * Conversely, the two remaining models exhibit large performance drops from removing the inspect framework: Grok 4 (58.5% to 42.2%) and DeepSeek V3.1 Terminus (35.7% to 31.3%). This demonstrates that some models (and in particular our leading model Grok 4 on subquestion performance) profit substantially from the tools provided by the inspect framework we set up.
>
> A detailed discussion will follow in the revision of our paper, planned for Nov 28th.
>
> **How will you mitigate grader load while preserving rigor?**
> We fully agree that the amount of work involved in grading any individual model answer would make it challenging or infeasible to perform the fine-grained proof-based evaluation of  every single newly released model. However, both the initial design of the benchmark, as well as some of the planned future developments are shaped to mitigate this limitation:
> Final-answer evaluation of subquestions can serve as a good proxy for model performance on the main question (correlation of 0.45, see also below) and allows an easy-to-evaluate method to obtain a relative ranking of the tested models. For the continuing development of the benchmark, the selection of further models for human grading will be guided by this measure. This ensures that the grading effort focuses on models close to the frontier for mathematical reasoning, which we consider to be the most interesting.
> As for grader motivation, we have observed very high turnouts: at the time of writing, 48 of the (now) 49 accepted questions have been graded by their authors. This can be explained by the intrinsic interest of the graders in the questions which they designed. On the other hand, the work is distributed on many shoulders, with the 48 questions having 30 authors. Apart from the intrinsic diversity this provides, it also means that the total amount of work per person stays very manageable.
> As demonstrated in recent work [1,2], LLMs have the potential to very reliably grade longform mathematical proofs. In future work, we plan to use our human grader data to develop, calibrate and test such automated grading systems that would allow more scalable evaluations.
>
>
> **Can you elaborate on the correlation between FA and proof scores?**
> Thank you for this suggestion! In the paper, we do already compare the correlation between the weighted subquestion score, and the 0-3 progress scores of the human graders, arriving at a correlation of 0.45 (Section 4.1). We chose to report this number since both quantities (weighted subquestion results and progress scores) allow some internal variation, compared to the harder binary criteria of getting full subquestion scores or a fully correct proof answer to the main question. Calculating this binary correlation one arrives at a value of 0.442, which we have now added to the paper.
>
> **Qualifications for community outreach**
> Several of the main authors are professional mathematicians themselves, engaged in active research and related activities (conferences) and have performed successful outreach events at conferences in the past (e.g. one with 25 participants who generated 40 question drafts, some of which were later refined and accepted in the benchmark).
>
>
> [1] https://arxiv.org/abs/2511.01846
>
> [2] https://arxiv.org/abs/2506.21621

---

### Official Review · Reviewer_KECi · 2025-11-05

**Soundness:** 3
**Presentation:** 3
**Contribution:** 1
**Rating:** 0
**Confidence:** 4

**Summary:**

The paper introduces a new benchmark that aims to evaluate LLMs performing research-level mathematics, with access to tools. The authors present a platform that they used to collect questions, as well as a question collection protocol. The benchmark currently contains 39 questions, and the authors state that more questions will be added. Several state-of-the-art LLMs are evaluated, and the authors describe various particularities of how the models reason, as well as promising results w.r.t. to solving rates (22% in case of GPT-5).

**Strengths:**

The question-creating pipeline is the strongest part of the paper. It is significantly more rigorous than what typical papers in the math benchmarking space achieve. The associate web codebase seems to be a genuine improvement that makes problem collection easier.

**Weaknesses:**

- **tiny benchmark size**: My main issue that prevents me from recommending this paper for acceptance is the size of the dataset: 39 questions, even if very carefully authored, manually reviewed (and accounting for 79 follow-up questions), fall significantly short of the number of questions needed to rigorously establish a benchmark that, "evaluates their [LLMs'] performance on research-level
tasks at the frontier of mathematical knowledge". To achieve this goal, many more data points need to be sampled, as each mathematical subdomain comes with its own issues (algebraic geometry exhibits different typical ways of reasoning than differential topology), which probably require a few dozen data points each to rigorously test.

- **the growth targets will not solve the problems** For comparison, papers typically used thousands of datapoints when autograding is possible (e.g. https://arxiv.org/html/2501.13766v1), and many hundreds when solely manual evaluation is employed (e.g., https://arxiv.org/abs/2301.13867).  Growing the dataset by the numbers the authors mention will not bring them to this level.

- **bias**: The word cloud (fig 12) shows a clear bias towards certain areas of mathematics; which is not surprising given the small sie of the benchmark.

- **open questions are not a plus**: "Of the 39 benchmark problems, authors characterize 7 as open research questions." I am unconvinced of the value of adding open research questions to a benchmark. The value of an open research question for a benchmark is only worthwhile if an LLM solves that question and thus highlights an awesome display of reasoning. Otherwise, the output of the LLM is hard to quantify (e.g., "meaningful but ultimately insufficient progress towards a solution")

- "Currently, IMProofBench consists of 39 problems developed in collaboration with over 23 mathematicians, with 30 more questions in the latest stages of the problem creation pipeline." I am not sure that the fact that "more is coming" (30 questions) is a good argument, as it is unclear when the questions will be devised.

- **benchmark is not repeatable:** The fact that the authors evaluated commercial LLMs makes the subsequent use of these questions unsuitable for these LLMs, as they have likely been ingested in the training data. Further, a lot of space is used for a verbal assessment of how the models perform. While this provides a snapshot, this information is already folklore, given the many posts on X/Twitter; at the same time, this information will soon be outdated with the advancements of LLMs. The value of a benchmark lies mostly in the fact of being able to rank models across time (e.g., the success of ImageNet) - and this benchmark fails at that task. The authors state that their benchmark will grow over time, but simply scaling it will not resolve these problems.

- **benchmark not public** In line with current trends, the authors do not make their benchmark public. This is claimed to preserve "intellectual property" of the creators (line 508). I am doubtful of this claim, as one could waive IP claims. This reduces reproducibility. I recommend the authors read on the FrontierMath debacle (https://www.reddit.com/r/math/comments/1iadcqw/the_frontiermath_scandal) that highlights the pitfalls of private benchmarks.

**Questions:**

- Were some existing benchmarks, such as https://arxiv.org/html/2501.13766v1, deliberately omitted from the Related Work section?

- "high-school or undergraduate-level mathematics (Balunovic et al., 2025; Frieder et al., 2023)" Some problems from the latter seem to be rather at the graduate level (at least in some universities' master curricula), as graduate-level textbooks feature in those benchmarks?

---

> ### Author Response · Authors · 2025-11-21
>
> We thank the reviewer for their review. However, we believe they significantly underestimate the difficulty of creating benchmark samples at this level and the value of our analysis. Further, we find several contradictory and factually incorrect statements in the review. Finally, we want to point out that 0 scores should only be used for works that do not meet very basic academic standards, or where scientific fraud has been detected. We thoroughly believe that our work meets these very basic requirements, and therefore urge the reviewer to reconsider their score. Below we address the particular points raised by the reviewer.
>
> **Did you deliberately omit [1] from related work?**
> Of course not. There are many mathematical benchmarks in existence, and the work pointed out by the reviewer is strictly less related to ours than works we do cite: it only covers undergraduate final-answer mathematical questions that are somewhat likely to be saturated by current LLMs (o1-mini obtained 68%). We have now included it in our discussion, but want to point out that it does not affect the value of our work in any way.
>
> **Do commercial providers train on data that is provided via API requests?**
> No, they do not. All major API providers explicitly state in their terms of use that API requests will not be used for training (see, for instance, [1]). Of course, we can make accusations that they do not abide by these terms (even though they are legally binding), but the same remark would generalize to**all**. Therefore, if the point raised by the reviewer would be accurate, then no benchmark scores should be trusted. Importantly, we never submit the answers in our API requests. Therefore, model providers would have to actively seek out our questions in the requests, attempt to answer them themselves, and then start training. This seems too malicious to be believable.
> Finally, we point out that for each model we add to our benchmark, we actively go through and read these terms of use. If they allow for training, then we do not add the model.
>
> **Why did you opt for private benchmarking?**
> Please see Q1 of our main reply, where we explain why the choice for a private benchmark is better in this case.
>
> Further, asking contributors to waive their IP claims and publicly release the problems they are currently working on, is unrealistic and will significantly limit the contributions made to the benchmark.
>
> Comparing the decision to have a private benchmark with the scandal regarding the FrontierMath benchmark is inaccurate, as this was caused by the private benchmark being executed very improperly, and not by it being a private benchmark. In particular, we do not, and never intend to, share our problems with third parties so they can be used for their own internal evaluations. As such, we will also never deceive our contributors about any such data sharing with third parties.
>
> **How does the small benchmark size affect the analysis?**
> Please see Q2 of our main reply.
>
> **How does the growth target of the dataset size compare to related work?**
> As mentioned in Q2 of our main reply, many benchmarks contain only a limited number of questions. IMOBench [2] is a recent work by Google Deepmind that contains a benchmark of 60 questions that are autograded by an LLM. Therefore, our benchmark size is not too dissimilar from prior work. Of course, there are works that evaluate many thousands of questions, but these generally trade-off size with noise.
>
> Further, while somewhat dismissed by the reviewer, the added benefit of having actual PhD researchers look at model answers provides many benefits. Our more detailed analysis requiring the grader to answer questions like usefulness is impossible to obtain through automated grading. Additionally, it allows us to perform our extensive qualitative analysis. This analysis allows us to go beyond aggregated scores, and is extremely important in current benchmarking efforts. It does not only allow us to tell when models fail, but also why, which is often a critical question to ask to further improve model performance.
>
> **What is the added value of the open questions?**
> Open questions still provide major benefits for our benchmark. First, due to our detailed analysis, answers to open questions can still reveal error patterns, which we also explicitly keep track of and analyze in Section 4.2. Additionally, these open questions can become important in the long run when models get sufficiently advanced to tackle these questions. As such, the inclusion of open questions indicates our long-term commitment to this task. Additionally, we want to point out that the UQ-Dataset [3] (cited in our related work) is built around evaluating unsolved questions.
>
> [1] https://platform.openai.com/docs/guides/your-data
>
> [2] https://imobench.github.io/
>
> [3] https://arxiv.org/abs/2508.17580

---

### Author Response · Authors · 2025-11-21
**Reply to all reviewers addressing shared concerns**

We thank the reviewers for their feedback. We are happy to hear that they found our problem creation pipeline rigorous (KECi, YB3b, Bjxz, H7Lc), our qualitative and quantitative analysis interesting (Bjxz, YB3b, xDmc), and our realistic evaluation environment thorough (Bjxz,xDmc). We have extracted three main concerns that are shared by most reviewers that we will answer in this main reply.

[Note on revision of our paper: the reviews raised excellent points, which we are addressing with several new experiments and evaluations (ablation tests for model scores given fewer or no tools, additional grading and reliability analysis, and various additional statistical tests). Due to additional prior time commitments of the main authors, we were not able to finish a revision incorporating all the new content this week, but will upload a full revision of the paper by November 28th at the latest. We apologize for the inconvenience.]

**Q1. Why did you opt for private benchmarking?**

We chose private over public benchmarking for three main reasons:
* **Contamination Prevention**: Each of our questions takes considerable effort to create, with a rigorous review policy in place to ensure their high quality. Therefore, they are not easily replaced by newer questions, making any form of contamination detrimental for performance measurements.
* **Web Search is Enabled**: We enable web search in our agentic setup, which we would need to disable if our benchmark questions were public. Indeed, once our questions appear online, there is no sufficiently rigorous way to prevent models from finding solutions.
* **Work by Contributors**: Contributors often submit questions from research projects they are currently working on. Since this work is not published, it is unreasonable to ask our contributors to allow us to publish these questions, as their own work could suffer from this.
Finally, we note that this decision is in line with other research-level benchmarks, such as FrontierMath [1] and MathScienceBench [2].

**Q2. How does the small scale affect your analysis?**

While the small scale somewhat limits our analysis, we find that this is the case for many benchmarks in the mathematical area. For instance, AIME 2025 and HMMT 2025 (two benchmarks reported on by almost all major model providers) only contain 30 questions each. USAMO 2025 and IMO 2025 (the former reported on in the release of Gemini-2.5-Pro Deep Think [3] and Grok-4 [4], and the latter used in evaluation by OpenAI and Gemini in July [5]) only contain six questions. Similarly, FrontierMath only contains 50 questions in its advanced Tier 4 subset and the more recent IMOBench [6] only contains 60 questions in its ProofBench (which is autograded). Therefore, the community seems to prefer a small set of very high-quality questions, rather than a larger set of more noisy samples. We take the same approach, implementing rigorous reviewing of submitted questions to ensure high quality, rather than going for the easier option that scales better.

Of course, the limited size of our dataset does limit our overall analysis. For instance, we cannot provide breakdowns by topic in research-level areas, since we do not have enough questions to do so. However, our analysis also provides a lot of value: we can actively compare models, we are fully independent from model providers, and while small, 39 questions still gives us a lot of signal to compare models with. For instance, the fact that model answers could help in approximately 50% of its replies, is a valuable and interesting conclusion.

We also note that, at the time of writing, we have added a total of 10 further questions to the benchmark, increasing the number of questions from 39 to 49. We are in the process of making an active push to review, evaluate and grade several further questions during the next 7 days.

**Q3. How does the current topical skew towards algebraic geometry affect your analysis?**

The reviewers correctly point that our current dataset is skewed towards algebraic geometry, mainly due to initial contributors being centered in that area. As our benchmark increases in size, we hope to rectify this skew by attracting more contributors from other areas as well.

However, we do not believe that this skew currently affects our analysis in any significant way. First, our dataset does contain questions from a range of other mathematical areas. Second, while quantitative results could somewhat differ between mathematical areas, our qualitative observations are very likely to generalise beyond our current dataset.

[1] https://epoch.ai/frontiermath

[2] https://math.science-bench.ai/

[3] https://blog.google/technology/google-deepmind/google-gemini-updates-io-2025/#flash-improvements

[4] https://x.ai/news/grok-4

[5] https://deepmind.google/blog/advanced-version-of-gemini-with-deep-think-officially-achieves-gold-medal-standard-at-the-international-mathematical-olympiad/

[6] https://imobench.github.io/

---

### Meta-Review · Area_Chair_kXme · 2026-01-06

**Summary:**

The submission is rejected. While the reviewers universally acknowledged the rigorous problem-creation pipeline, the novel proof-centric evaluation strategy, and the high-fidelity agentic environment, a consensus regarding fundamental methodological barriers precludes acceptance. The primary barrier dictating this outcome is the tension between the proposed private, non-repeatable dataset structure and the scientific necessity for reproducibility, auditability, and scale. Although Reviewer KECi assigned a score of 0, an outlier that has been statistically discounted for the final decision, their qualitative critiques regarding the extremely small benchmark size, topical bias, and the opacity of a closed evaluation ecosystem were echoed to varying degrees by the other reviewers. The committee finds that while the motivation is strong, the execution currently lacks the transparency required for a standard benchmark paper.

**Reviewer Concerns:**

The authors provided a robust rebuttal that successfully improved the completeness of the work. Notably, the inclusion of specific ablation results (isolating tool effects versus code interpreters), the addition of statistical error bars, and the clarification of the work's independence from frontier mathematics benchmarks addressed significant technical gaps. Furthermore, the commitment to releasing code, sample problems, and detailed grader guidelines demonstrates a good-faith effort to improve transparency. However, the soundness of the core contribution remains constrained by the architectural choices of the benchmark itself. Despite the authors' reasonable defense of a private design to prevent contamination, the absence of a frozen, public test split renders external verification impossible. The reliance on human graders for a small-scale dataset creates a bottleneck that limits statistical power, and the proposed 'monthly snapshots' do not sufficiently resolve the community's need for a stable, reproducible reference point.

**Reviewer Scores:**

Based on the dialectical exchange, the scoring landscape would likely stabilize but not cross the acceptance threshold. Reviewer KECi's score would ideally adjust from 0 to 2; while the initial score was excessively punitive, the rebuttal's defense of the private design did not resolve the reviewer's fundamental concerns regarding scale and public access, effectively capping the score. Reviewers xDmc and YB3b would likely maintain their scores of 4. Both reviewers acknowledged the substantive technical progress regarding tool ablations and privacy rationales, yet they correctly identified that the private nature of the data and the small 'N' count are structural limitations that a rebuttal cannot retroactively fix. Finally, Reviewers Bjxz and H7Lc would likely hold steady at 6. The introduction of inter-rater reliability plans and the clarification of the 'model-as-service' framing solidified their positive assessments, making the residual concerns about tool parity and dataset evolution manageable within their specific evaluation frameworks, though not sufficient to advocate for a higher score or sway the broader consensus.

---

### Decision · Program_Chairs · 2026-01-26

Reject